# RPNT: Robust Pre-trained Neural Transformer - A Pathway for Generalized Motor Decoding

## Abstract

Brain decoding aims to interpret and translate neural activity into behaviors. As such, it is imperative that decoding models are able to generalize across variations, such as recordings from different brain sites, distinct sessions, different types of behavior, and a variety of subjects. Current models can only partially address these challenges and warrant the development of pretrained neural transformer models capable to adapt and generalize. In this work, we propose **RPNT** - **R**obust **P**retrained **N**eural **T**ransformer, designed to achieve robust generalization through pretraining, which in turn enables effective finetuning given a downstream task. To achieve the proposed architecture of RPNT, we undertook an investigation to determine which building blocks will be suitable for neural spike activity modeling, since components from transformer models developed for other modalities do not transfer directly to neural data. In particular, RPNT unique components include 1) Multidimensional rotary positional embedding (MRoPE) to aggregate experimental metadata such as site coordinates, session name and behavior types; 2) Context-based attention mechanism via convolution kernels operating on global attention to learn local temporal structures for handling non-stationarity of neural population activity; 3) Robust self-supervised learning (SSL) objective with uniform causal masking strategies and contrastive representations. We pretrained two separate versions of RPNT on distinct datasets a) Multi-session, multi-task, and multi-subject microelectrode benchmark; b) Multi-site recordings using high-density Neuropixel 1.0 probes. The datasets include recordings from the dorsal premotor cortex (PMd) and from the primary motor cortex (M1) regions of non-human primates (NHPs) as they performed reaching tasks. After pretraining, we evaluated the generalization of RPNT in cross-session, cross-type, cross-subject, and cross-site downstream behavior decoding tasks. Our results show that RPNT consistently achieves and surpasses the decoding performance of existing decoding models in all tasks. Our ablation and sweeping analysis demonstrate the necessity and robustness of the proposed novel components.

## 1 Introduction

The transformer architecture (Vaswani et al., 2017), along with pretraining, has greatly shifted data modeling paradigms in the field of natural language processing and subsequently in a variety of fields. Indeed, transformer models such as BERT (Devlin et al., 2019), GPT (Radford et al., 2019), and Vision Transformers (Dosovitskiy et al., 2020) demonstrated that large-scale pretraining on diverse data followed by downstream task-specific finetuning yields more enhanced performance on a variety of related tasks compared to task-specific models. However, for neural data, which contains session-based nonstationary spatiotemporal structure, neural models have yet to utilize such strategies in the full extent. Typical variations in neural data are due to recordings from different brain sites, distinct sessions, different types of behavior, and a variety of subjects.

Efforts have been made recently. Methods such as Latent Factor Analysis via Dynamical Systems (LFADS) (Pandarinath et al., 2018) pioneered multi-session pretraining. Later, transformer-based models such as self-supervised Neural Data Transformer and its variants (NDT (Ye & Pandarinath, 2021), NDT2 (Ye et al., 2023), NDT3 (Ye et al., 2025)) and supervised pretraining models such as

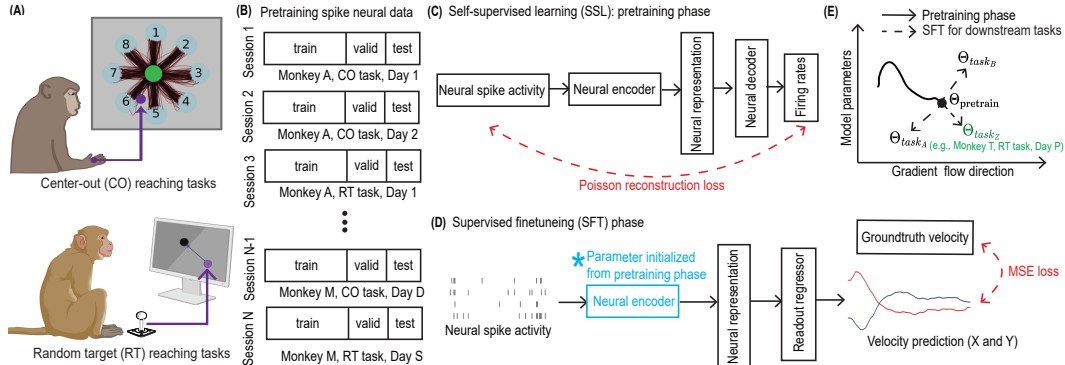

Figure 1: Overall illustration of the pretraining and finetuning workflow for generalized motor decoding. (A) Experimental setup for data collection while NHPs performed reaching tasks. (B) Preparation of pretaining data. (C) and (D) overall schemes for SSL and SFT, respectively. (E) Illustration of model adaptation during pretraining and finetuning.

POYO (Azabou et al., 2023), PoYO+ (Azabou et al., 2024), PoSSM (Ryoo et al., 2025) showed the benefits of pretraining and finetuning in the neural decoding workflow.

While these efforts have adapted pretraining strategies to neural datasets with larger samples of neural activity (i.e., neural population and neural dynamics models) and more diverse samples (distinct recording sessions and subjects), robust generalization via explainable mechanisms for possible variations in neural data remains a challenge. In particular, several key aspects of variation in neural data have yet to be resolved. First, model generalization capabilities remain limited when applied to unseen brain sites or different recording configurations (Jude et al., 2022; Karpowicz et al., 2024; Le et al., 2025). Second, neural signals drift over time (Chestek et al., 2011). Thus, building powerful mechanisms that can handle neural non-stationarity remains unclear. Third, existing approaches typically emphasize training objectives that ultimately improve performance in decoding tasks by denoising neural signals, but may be unable to learn representations that are informative of the underlying causal dynamics that give rise to neural activity (Lu et al., 2025).

To address the above limitations, we introduce RPNT - a neural transformer model that incorporates robust pre-training enhancements. RPNT contains novel components that are a result of an investigation aimed at designing a transformer model specifically for neural activity and its variations. In particular, the key novel components of RPNT are: 1) Multidimensional rotary positional embedding (MRoPE) to aggregate experimental metadata such as site coordinates, session name and behavior types; 2) Context-based attention mechanism via convolution kernels operating on the global attention to learn local temporal structures for handling the non-stationarity of neural activity; 3) Robust self-supervised learning (SSL) objective with uniform causal masking strategies and contrastive representations.

For validation of RPNT, we implemented a standard pretraining and finetuning workflow (see Figure 1). With these workflows, we pretrained two separate versions of RPNT on two datasets which include different modalities of the recorded neural activity: a) Multi-session, multi-task, and multi-subject microelectrode benchmark data (Perich et al., 2018); b) Multi-site recordings using high-density Neuropixel 1.0 probes. Both datasets include recordings from the (PMd) and the (M1) regions of nonhuman primates (NHPs) as they performed center-out or random target reaching tasks. After pretraining, we evaluated the generalization of RPNT in cross-session, cross-type, cross-subject, and cross-site downstream behavior decoding tasks. Our results showed that RPNT consistently achieved and surpassed the decoding performance of existing decoding models in all tasks. Our ablation and sweeping analysis demonstrated the necessity and robustness of the proposed novel components. Furthermore, the model's learned attention map provided data-driven insights into the underlying structure of neural encoding of movement variables. Our approach has implications for future large-scale, robust neural transformer pretraining for neural decoding applications such as brain-computer interfaces (BCIs) and data-driven discovery for neuroscience. To this end, our major contributions are listed:

- **Model architecture**: We propose a multidimensional rotary positional embedding (MRoPE) to aggregate experimental metadata for decoding generalization. We introduce a context-based local attention mechanism to handle the neural non-stationarity.

- **Pretraining strategy**: We pretrain RPNT using a robust SSL objective via uniform causal masking strategies and contrastive representations.

- **Generalized Decoding**: We demonstrate superior decoding performance for 1) cross-session, cross-behavior types, and cross-subject scenarios in the microelecotrode recording benchmark; 2) cross-site scenario in a new Neuropixels dataset.

## 2 RELATED WORK

**Neural Population Modeling** Classical approaches to neural population analysis proposed to leverage dimensionality reduction to extract low-dimensional structure from high-dimensional recordings. Methods such as PCA (Cunningham & Yu, 2014) and Factor Analysis (Santhanam et al., 2009) were applied to neural activity and showed identification of dominant modes of variance. Since such methods rely on the assumption of static and linear relationships, extensions have been developed, such as a probabilistic geometric PCA with application to neural data (Hsieh & Shanechi, 2025) and Canonical Correlation Analysis to cross-region interactions (Semedo et al., 2019). While extending classical dimension reduction approaches, such methodologies remain limited to pairwise correlations. As an alternative, dynamical systems approaches such as GPFA (Yu et al., 2008) and LFADS (Pandarinath et al., 2018) have been introduced to model the temporal evolution of latent states. While such models are applicable to ubiquitous interpretation of neural population data, their main applications are focused on offline neural denoising analysis rather than neural decoding.

**Motor Decoding** Motor decoding methods aim to reconstruct intended movements or kinematic variables from neural population activity. Classical approaches, such as Wiener (Van Drongelen, 2018) and Kalman filters (Wu et al., 2002; Orsborn et al., 2014), provide real-time decoding with computational efficiency, however typically assume linear dynamics and Gaussian noise. Bayesian decoders (Wu et al., 2006; Shanechi et al., 2016) incorporate prior information about movement statistics, improving robustness but remaining limited by parametric assumptions. Deep learning approaches have been developed as well, and these progressed from MLPs (Glaser et al., 2020) to RNN architectures (Mante et al., 2013; Sani et al., 2024) aimed to capture temporal dependencies in neural dynamics and thus perform effective decoding. Recent works explore Transformer-based decoders (Ye & Pandarinath, 2021; Le & Shlizerman, 2022) that leverage self-attention to model long-range dependencies across neural populations. However, these methods predominantly train from scratch on limited task-specific data, leading to poor generalization across sessions, subjects, and experimental conditions.

**Pretraining of Neural Models** Efforts have been made to use large-scale pretraining for neural data analysis with various model architectures. Before the era of the transformer, RNN-based models were explored for multi-session pretraining, such as MRNN (Sussillo et al., 2016) and LFADS (Pandarinath et al., 2018). Then, transformer architecture-based models, i.e., Neural Data Transformer (NDT) (Ye & Pandarinath, 2021) and Pre-training On manY neurOns (PoYo) (Azabou et al., 2023) work and subsequent works represent additional developments toward pretraining of neural models. In particular, NDT (Ye & Pandarinath, 2021) introduced a transformer architecture for spike train modeling, while NDT2 (Ye et al., 2023) demonstrated multi-context pretraining across sessions, subjects, and tasks using a spatiotemporal transformer. NDT3 (Ye et al., 2025) scaled it to hundreds of datasets and showed emergent zero-shot/few-shot capabilities. Concurrently, supervised pretraining models such as POYO (Azabou et al., 2023), PoYO+ (Azabou et al., 2024), PoSSM (Ryoo et al., 2025) took a complementary approach of leveraging learnable neural token representation to realize scalable pretraining across multiple subjects. Additional approaches have been proposed as well. Models such as Population Transformer (PopT) used standard transformer encoder architecture but with novel ensemble-wise and channel-wise discrimination tasks to train their model (Chau et al., 2025). Further, BrainBERT (Wang et al., 2023) proposed to adapt language model architectures directly, treating neural signals as text sequences. Although effective, these transformer models lack components for dealing with the non-stationarity and recording configurations in session-based neural spike activity.

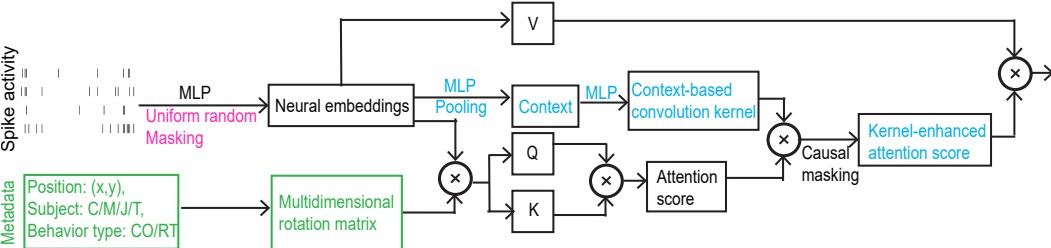

Figure 2: A schematic of components in RPNT. Components in black indicate standard transformer signal flow (i.e, no masking and standard attention mechanism). Our novel proposed components include MRoPE (green), context-based attention (cyan), and uniform random masking strategy (pink). MRoPE incorporates experimental metadata into the transformer positional embeddings. The context-based attention captures local temporal structures. The input spike activity is randomly masked according to a uniform distribution for causal masked neural modeling. The masked input spike activity is projected into neural embeddings, forming Q, K, and V representations. MRoPe is applied to get the rotary queries and keys. Attention scores are modulated by context-based convolutional kernels and combined with causal masking before aggregating with values.

**Attention Mechanisms and Positional Embedding for Non-stationary Data:** The originally developed self-attention mechanism in (Vaswani et al., 2017) assumes stationary input distributions, which may not be suitable for neural recordings where signal characteristics drift over time (Chestek et al., 2011). Adaptive attention mechanisms have been proposed for domains with nonstationary data, such as video (Chen et al., 2021). These mechanisms are based on learnable kernels to modulate the attention based on local context. Further, in (Zhu et al., 2021) a sliding window attention was introduced to capture local structure, and in (Yang et al., 2016) a hierarchical attention across multiple timescales was employed. Non-stationary Transformers (Liu et al., 2022) introduce series stationary and nonstationary attention to handle distribution shifts in time-series forecasting. FEDformer (Zhou et al., 2022) employs frequency-enhanced decomposed architectures to capture both seasonal and trend patterns in nonstationary signals. Crossformer (Zhang & Yan, 2023) utilizes dimension-segment-wise attention to capture cross-time and cross-dimension dependencies.

Beyond the attention mechanism, positional encoding also presents challenges due to the unique nature of neural data. Particularly, positional encoding is expected to take into consideration the joint recording configurations and time. While standard sinusoidal encodings (Vaswani et al., 2017) and learned positional embeddings (Devlin et al., 2019) are generally applicable, recent positional encodings could offer advantages for modeling relative position exclusive to one of the dimensions, space or time. In particular, RoPE (Su et al., 2024) allows modeling of relative positions exclusively in the temporal dimension. Recent extensions of RoPE to multiple dimensions have shown promise in handling complex positional relationships beyond simple sequential ordering. The Qwen2 (Wang et al., 2024b) demonstrates effective multi-dimensional RoPE for handling both sequence position and other modality dimensions. Here, we embrace RoPE for neural activity (Azabou et al., 2023) and use the extended MRoPE (Wang et al., 2024b) for capturing recording configurations.

## 3 METHODS

### 3.1 MROPE: MULTI-DIMENSIONAL ROTARY POSITIONAL EMBEDDING

Since neural recordings vary, it is imperative that each neural recording configuration should be explicitly modeled within the transformer positional embedding. Such embedding is beyond the temporal positional embedding. For example, in multiple brain sites recordings, an effective positional embedding would be one that can model both spatial site locations and temporal relationships for the generalization to unseen site configurations. A plausible candidate for such embedding that we found is Rotary Positional Embedding (RoPE) (Su et al., 2024), which we extend here to multiple dimensions, i.e., MRoPE, where $M$ stands for the configuration dimensions, to aggregate experimental metadata. In the Neuropixel dataset, $M = 3D$, and it includes two-dimensional spa-

tial coordinates (denoted by $(x_s, y_s)$) and time $t$. To construct the rotational matrix, we partition the model dimension ($d_{\text{model}}$) into three groups independently ($d_{\text{coord}} = \frac{d_{\text{model}}}{3}$) to represent dimensions each for $x$-coordinate, $y$-coordinate, and temporal position. For each group, we define rotation frequencies:

$$\theta_i^{(x)} = \frac{1}{5000^{2i/d_{\text{coord}}}}, \quad \theta_i^{(y)} = \frac{1}{5000^{2i/d_{\text{coord}}}}, \quad \theta_i^{(t)} = \frac{1}{10000^{2i/d_{\text{coord}}}}, \tag{1}$$

where $i \in [0, d_{\text{coord}}/2)$. Spatial dimensions use lower frequency ($f = 5000$) for coherence across nearby brain sites, while temporal dimensions use standard RoPE frequency ($f = 10000$) for fine-grained temporal resolution.

$$\mathbf{R}_{3D}^d = \begin{bmatrix} \mathbf{R}_x(x) & \mathbf{0} & \mathbf{0} \\ \mathbf{0} & \mathbf{R}_y(y) & \mathbf{0} \\ \mathbf{0} & \mathbf{0} & \mathbf{R}_t(t) \end{bmatrix} \in \mathbb{R}^{d \times d} \tag{2}$$

**Rotation Operation:** In contrast to traditional positional encodings added to embeddings, MRoPE works similarly to RoPE, by applying rotations directly to query and key vectors during attention computation. For 3D position $(x_s, y_s, t)$, we construct a block-diagonal rotation matrix $\mathbf{R}_{3D}(x_s, y_s, t)$(see equation 2) that independently rotates dimension pairs. The attention score between location $i$ and $j$ follows $\mathbf{q}_i^T \mathbf{k}_j = \mathbf{q}^T \mathbf{R}_{3D}^T(x_i, y_i, t_i)\mathbf{R}_{3D}(x_j, y_j, t_j)\mathbf{k} = f(x_j - x_i, y_j - y_i, t_j - t_i)$, which preserves the key relative position property. Such operations enable zero-shot generalization to arbitrary brain site recording configurations while maintaining rotational invariance (also see the appendix).

MRoPE is designed for arbitrary cases where the recording configurations encounter different subjects, recording times, locations, and behavior types (see Figure 2). Specifically, we can partition the model dimension $d_{\text{model}}$ into $M$ groups depending on the number of recording configurations, i.e., $\theta_i^{(1)} = \frac{1}{f_1^{2i/d_1}}, \quad \cdots \quad, \theta_i^{(\mathcal{M})} = \frac{1}{f_M^{2i/d_M}}$, where $d_1, \cdots, d_M$ are the dimensions for each group that in total contribute to $d_{\text{model}}$. The $f_1, \cdots, f_M$ are the corresponding frequencies. Indeed, for the public benchmark, we set MRoPE with $M = 4D$ to represent behavior types, subjects, recording times, and temporal position. We used an equal dimension partition ($d_m = \frac{d_{model}}{4}$) and set frequencies with 10, 100, 1000, and 10000, respectively (see the appendix for further details).

## 3.2 CONTEXT-BASED ATTENTION MECHANISM

To address non-stationary neural activity (Chestek et al., 2011), we introduce a context-based attention mechanism via learnable convolution kernels (Gulati et al., 2020) operating on global attention to learn the local temporal structure. Such attention is achieved through the incorporation of history and pooling into attention. We illustrate the attention steps in Figure 2 (blue) and describe them below.

**Context Generation:** Given a batch neural input $\mathbf{x} \in \mathbb{R}^{B \times T \times D}$ and historical data $\mathbf{H}_{\text{hist}} \in \mathbb{R}^{B \times T_{\text{hist}} \times D}$ containing only past timesteps, we generate a context vector through attention pooling:

$$\mathbf{c} = \text{AttentionPool}(\text{MLP}(\mathbf{H}_{\text{hist}})) \in \mathbb{R}^{B \times D_{\text{context}}} \tag{3}$$

**Dynamic Kernel Generation:** We parameterize the context vector to generate 2D convolution kernels for each attention head:

$$\mathbf{K}_h = \text{reshape}(\text{softmax}(\text{MLP}_{\text{kernel}}(\mathbf{c})), [K_1, K_2]), \tag{4}$$

where $h \in [1, H]$ indexes attention heads and $(K_1, K_2)$ are hyperparameters for convolution kernel dimensions.

**Kernel-Enhanced Attention:** As the last step, we apply a 2D convolution kernel after calculating the standard attention matrix for better capturing the local attention relationship:

$$\mathbf{A}_{\text{kernel}} = \text{Conv2D}(\frac{\mathbf{Q}\mathbf{K}^T}{\sqrt{d}}, \mathbf{K}_h) \odot \mathbf{M}_{\text{causal}}, \tag{5}$$

where $\mathbf{M}_{\text{causal}}$ is the causal masking ($\mathbf{M}_{\text{causal}}[i, j] = \Vdash[j \leq i]$). This design enables each head to learn specialized temporal dependencies for handling non-stationarity (see Figure 2).

### 3.3 NEURAL TRANSFORMER ARCHITECTURE

With the above two components, we compose RPNT's neural transformer encoder, where we decouple temporal and cross-site processing (for the neuropixel dataset). First, we build a **temporal encoder**. Given neural spike activity $\mathbf{X} \in \mathbb{R}^{B \times S \times T \times N}$ (batch, sites, time, neurons) and corresponding experimental metadata, we first embed each recording site $\mathbf{H}_s^{(0)} = \text{Linear}(\mathbf{X}_{:,s,:,:})$. Then, we build a temporal encoder that consists of $L$ transformer layers with MRoPE and context-based attention

$$\mathbf{H}_s^{(l+1)} = \mathbf{H}_s^{(l)} + \text{ContextAttn}(\text{LN}(\mathbf{H}_s^{(l)}), \text{MRoPE}(\text{configurations})). \tag{6}$$

Most of the neural representation learning and neural population modeling can be done via the above powerful temporal encoder by treating the number of recording sites $S = 1$, i.e., repeated neural measurements across sessions. Further, if it is desired to model the cross-site neural dynamics beyond the temporal encoding, such as sampled neuropixel recordings for cross-site activity, we propose to concatenate a **spatial encoder** right after the temporal encoder. Spatial encoder will model cross-site interactions through timestep-wise processing, naturally preserving causality. At each timestep $t$, we follow the standard multihead attention (MHA)

$$\mathbf{Z}_{:,:,t,:}^{(l+1)} = \mathbf{Z}_{:,:,t,:}^{(l)} + \text{MultiheadAttn}(\text{LN}(\mathbf{Z}_{:,:,t,:}^{(l)})), \tag{7}$$

where $\mathbf{Z}^{(0)} = \mathbf{H}^{(L)}$ from the last layer of temporal encoder. The MHA weights can provide interpretable cross-site functional connectivity. In our experiments, we use the microelectrode benchmark (Perich et al., 2018) mainly for the purpose of investigating and validating the power of the temporal encoder and the neuropixel dataset to further show the attention map visualization for data-driven discovery by the spatial encoder.

### 3.4 CAUSAL MASKED NEURAL MODELING

With the neural transformer encoder setup, we introduce a pretraining framework tailored for neural spike data that robustly performs self-supervised learning (SSL) using masked causal Poisson reconstruction loss with uniform random masking and cross-site contrastive loss.

**Uniform Random Masking Strategy.** Unlike fixed masking strategies, which required careful tuning in prior work (Srivastava et al., 2014; Ye et al., 2025; Zhang et al., 2024; He et al., 2022), we apply masking both in space (neurons) and time according to the uniform distribution, i.e., $p_m \sim \mathcal{U}(0, 1)$, at each batch. This choice is motivated by works of Azabou et al. (2023) and (Le et al., 2025), where in Azabou et al. (2023) it was shown that dynamic dropout augmented the decoding performance, and in a recent work of (Le et al., 2025) it was shown that uniform neuron sampling strategy can improve the supervised motor decoding performance. These findings led us to further extend masking to both time and neuron during SSL pretraining. The stochastic approach that we propose eliminates the masking rate hyperparameter while ensuring the model encounters diverse reconstruction difficulties during the pretraining phase, i.e., $\mathbf{M}_{:,:,t,n} \sim \text{Bernoulli}(1 - p_m)$.

**Self-Supervised Learning Objective.** Distinct from existing denoising-based transformer work, our approach maintains causality, i.e., reconstruction at time $t$ depends only on unmasked inputs from $t' \leq t$. This transforms the standard MAE objective He et al. (2022) into a predictive task aligned with the autoregressive nature of neural mechanisms (see Appendix for the comparison). For masked position $(t, n)$, RPNT reconstructs it with respect to

$$\hat{x}_{:,t,n} = f_\theta(\{\mathbf{x}_{:,t',n'} : (:, t', n') \notin \mathcal{M} \wedge t' \leq t\}). \tag{8}$$

As neural spike counts follow a Poisson distribution, we use the Poisson reconstruction loss

$$\mathcal{L}_{\text{recon}} = \sum_{(:,t,n) \in \mathcal{M}} \left[ \hat{\lambda}_{:,t,n} - x_{:,t,n} \log(\hat{\lambda}_{:,t,n} + \epsilon) \right], \tag{9}$$

where $\hat{\lambda}_{:,t,n} = \text{Decoder}(\mathbf{Z}_{:,t,d})$ and $\epsilon = 10^{-8}$ provides numerical stability (see decoder design in appendix). Further, an auxiliary objective encourages learning site-invariant representations (see the appendix for the formulation and ablation study). The combined objective $\mathcal{L} = \mathcal{L}_{\text{recon}} + \mu \mathcal{L}_{\text{contrast}}$ balances reconstruction objective with robust representation learning.

### 3.5 Downstream Evaluation Framework

We evaluate the generalizability of RPNT on downstream behavior decoding tasks across four scenarios (cross-session, cross-type, cross-subject, and cross-site). Specifically, we build lightweight task-specific heads following the last layer of the pretrained temporal encoder of RPNT

$$\hat{\mathbf{y}}_t = \text{MLP}_{\text{task}}(\mathbf{H}_{:,t}), \tag{10}$$

where $\mathbf{H}_{:,t}$ is the temporal encoder outputs across at time $t$. We use mean-squared error (MSE) for SFT on velocity regression tasks and evaluate using $R^2$ metric.

Further, we visualize functional connectivity (FC) based on the spatial attention map across time. The spatial encoder's cross-site attention weights directly encode functional relationships. We extract the connectivity matrix $\mathbf{C} \in \mathbb{R}^{T \times S \times S}$ by averaging attention weights across layers, i.e., $\mathbf{C}_{i \to j}(\text{attention}) = \frac{1}{L} \sum_{l=1}^{L} \text{Attention}_{i \to j}^{(l)}$. The visualization results were shown in the appendix.

## 4 Experiments

### 4.1 Datasets and experimental setup

**Datasets** We evaluated the downstream generalization of the RPNT model under three out of four scenarios of variation (cross-session, cross-task, and cross-subject scenarios) on the public micro-electrode benchmark (Perich et al., 2018). For the fourth scenario of cross-site variation, we performed experiments with a Neuropixel dataset. The (Perich et al., 2018) dataset contains electro-physiology and behavioral data from 4 rhesus macaques (subjects C, J, M, and T) performing either a Center-Out reaching task (denoted as CO) or a continuous random target acquisition task (denoted as RT). Neural activities were recorded from Utah arrays in the M1 or PMd region. This dataset contained 111 sessions, spanning 43 recording hours. Both raw data and preprocessed data can be accessed from the Python package *brainsets* provided by (Azabou et al., 2023). In the Neuropixel dataset, a male rhesus macaque (named subject B) was implanted with a multi-modal chamber over the frontal motor cortex. The subject was trained to perform the CO reaching tasks. At each experimental session, a single neuropixel probe was inserted in the chamber to acquire specific recording-site neural activity. The dataset is comprised of 17 recordings at different sites distributed across the PMd and M1 regions across experiment sessions spanning approximately one year.

**Experimental setup** We have conducted experiments of RPNT function and validation first focused on self-supervised pretraining of the RPNT model and then evaluation and comparison of RPNT downstream decoding generalization with baselines through supervised finetuning (SFT) on test recordings (see Figure 1E for model adaptation). For benchmark (Perich et al., 2018), we followed the baseline setup according to (Azabou et al., 2023; Ryoo et al., 2025), where we used the train session data in subjects C, J, and M for pretraining and tested with the rest of the sessions for downstream evaluations. Specifically, our decoding evaluation included the following three cases: 1) New center-out sessions from Monkey C (denoted as C-CO) for cross-session evaluation; 2) New center-out sessions from Monkey T (denoted as T-CO) for cross-subject evaluation; 3) New random target sessions from Monkey T (denoted as T-RT) for cross-subject and cross-tasks evaluation. Notably, more than $80\%$ of sessions in the benchmark are CO tasks. Thus, the T-RT evaluation is expected to be the most challenging task, since it requires the model to adjust to both subject and task distribution shift. We followed the released code for building the train/valid/test datasets according to the Python package *torch brain* (Azabou et al., 2023). For the neuropixel dataset, we used 16 (S1-S16) sites of recording neural activity (with 0.8/0.1/0.1 for train/val/test split) to pretrain the RPNT model, then used the remaining site (S17) (with 0.2/0.3/0.5 for train/val/test split) for FS cross-site evaluation. Further, as the CO task was highly structured, we followed the convention by only evaluating the decoding performance at the reaching period on the successful trials (Ryoo et al., 2025). We used $R^2$ to quantify the decoding performance between the predicted and the ground-truth velocity behaviors. The baseline comparison models are described in the appendix.

| | Method | Cross-Session (C-CO) | Cross-Subject (T-CO) | Cross-Task (T-RT) |
|---|---|---|---|---|
| Scratch | Wiener filter | $0.8712 \pm 0.0137$ | $0.6597 \pm 0.0392$ | $0.5942 \pm 0.0564$ |
| | MLP $^\dagger$ | $0.9210 \pm 0.0010$ | $0.7976 \pm 0.0220$ | $0.7007 \pm 0.0774$ |
| | S4D $^\dagger$ | $0.9381 \pm 0.0083$ | $0.8526 \pm 0.0243$ | $0.7145 \pm 0.0671$ |
| | Mamba $^\dagger$ | $0.9287 \pm 0.0034$ | $0.7692 \pm 0.0235$ | $0.6694 \pm 0.1220$ |
| | GRU $^\dagger$ | $0.9376 \pm 0.0036$ | $0.8453 \pm 0.0200$ | $0.7279 \pm 0.0679$ |
| | POYO-SS $^\dagger$ | $0.9427 \pm 0.0019$ | $0.8705 \pm 0.0193$ | $0.7156 \pm 0.0966$ |
| | POSSM-S4D-SS $^\dagger$ | $0.9515 \pm 0.0021$ | $0.8838 \pm 0.0171$ | $0.7505 \pm 0.0735$ |
| | POSSM-Mamba-SS $^\dagger$ | $0.9550 \pm 0.0003$ | $0.8747 \pm 0.0173$ | $0.7418 \pm 0.0790$ |
| | POSSM-GRU-SS $^\dagger$ | $0.9549 \pm 0.0012$ | $0.8863 \pm 0.0222$ | $0.7687 \pm 0.0669$ |
| | **RPNT** | $\mathbf{0.9647} \pm 0.0026$ | $\mathbf{0.9103} \pm 0.0182$ | $\mathbf{0.8356} \pm 0.0914$ |
| Pretrained | NDT-2 $^\dagger$ | $0.8507 \pm 0.0110$ | $0.6549 \pm 0.0290$ | $0.5903 \pm 0.1430$ |
| | POYO-1 $^\dagger$ | $0.9611 \pm 0.0035$ | $0.8859 \pm 0.0275$ | $0.7591 \pm 0.0770$ |
| | o-POSSM-S4D $^\dagger$ | $0.9618 \pm 0.0007$ | $0.9069 \pm 0.0120$ | $0.7584 \pm 0.0637$ |
| | o-POSSM-Mamba $^\dagger$ | $0.9574 \pm 0.0016$ | $0.9011 \pm 0.0148$ | $0.7621 \pm 0.0765$ |
| | o-POSSM-GRU $^\dagger$ | $0.9587 \pm 0.0052$ | $0.9021 \pm 0.0241$ | $0.7717 \pm 0.0595$ |
| | **RPNT** (FS-SFT) | $\mathbf{0.9801} \pm 0.0060$ | $\mathbf{0.9431} \pm 0.0103$ | $\mathbf{0.8515} \pm 0.1071$ |
| | **RPNT** (Full-SFT) | $\mathbf{0.9894} \pm 0.0037$ | $\mathbf{0.9626} \pm 0.0059$ | $\mathbf{0.8778} \pm 0.1005$ |

$^\dagger$ indicate results are copied from the POSSM (Ryoo et al., 2025) paper

Table 1: Velocity decoding performance ($R^2$) comparison across three generalization scenarios (C-CO, T-CO, and T-RT) on the public benchmark. RPNT (FS-SFT) used few-shot finetuning, whereas RPNT (Full-SFT) used all available downstream sessions. The mean and standard deviation were calculated across all sessions. Bold numbers indicated that the SOTA performance in each scenario. The references for baseline models are listed in the appendix.

### 4.2 CROSS-SESSION, CROSS-SUBJECT, AND CROSS-TASK EVALUATION RESULTS ON PUBLIC BENCHMARK

We compare the velocity decoding performance ($R^2$) across three generalization scenarios on the benchmark in Table 1, where RPNT was evaluated with two types of baselines and training: 1) models trained from scratch on single downstream sessions and 2) pretrained models finetuned with different amounts of downstream sessions.

**Training from scratch.** In this regime (top segment of Table 1), compared models are trained for each downstream session only. RPNT achieved the best performance across all three scenarios. Specifically, RPNT obtained $R^2$ scores (mean $\pm$ std) of $0.9647 \pm 0.0026$, $0.9103 \pm 0.0182$, and $0.8356 \pm 0.0914$ in C-CO, T-CO, and T-RT evaluations, respectively. These results show enhancement with respect to existing decoding models, with particularly notable achievement in the challenging T-RT scenario ($\approx 7\%$ increase from second second-best baseline POSSM-GRU).

**Pretrained models.** We next investigated the benefits of pretraining and a downstream supervised finetuning (SFT) approach. As a self-supervised pretraining model, RPNT finetuning can operate in two regimes: a) Finetuning on the full training dataset of the downstream task (Full-SFT) and b) More restrictive few-shot finetuning, where finetuning is performed on a single downstream session (FS-SFT). RPNT was tested in these two regimes and compared with baselines in the FS-SFT regimes. Results (bottom segment of Table 1) indicate that both RPNT(Full-SFT) and RPNT(FS-SFT) outperformed previous baselines in the FS-SFT regime. Notably, the more challenging regime of testing RPNT, RPNT(FS-SFT), achieved average $R^2$ scores of 0.9801, 0.9431, and 0.8515 for three tasks, in contrast to existing state-of-the-art POYO and POSSM baselines in the FS-SFT regime, which had access to behavior labels for all sessions during the pretraining phase. In the comparable label usage of Full-SFT, RPNT $R^2$ was further enhanced compared to the results on the FS-SFT regime and reached $0.9894 \pm 0.0037$ (C-CO), $0.9626 \pm 0.0059$ (T-CO), and $0.8778 \pm 0.1005$ (T-RT). Due to these results, and the potential of the FS-SFT regime to require more efficient finetuning in latter experiments of section 4.3 and section 4.4, we fixed RPNT in the regime of FS-SFT. As we report in the results below, RPNT surpasses the performance of existing models, indicating the efficiency and robustness of the RPNT in decoding task generalization. With respect to RPNT variants, we note that our results indicate that RPNT that is pretrained outperforms RPNT that is

trained from scratch, e.g., Pretrained RPNT (Full-SFT): 0.8778 vs. RPNT from scratch 0.8356 in the T-RT task, indicating the benefits of the self-supervised pretraining strategy for robust neural representation learning.

### 4.3 CROSS-SITE EVALUATION RESULT ON THE NEUROPIXEL DATASET

| Method | Cross-Site (B-CS) |
|---|---|
| Wiener filter | $0.3462 \pm 0.0710$ |
| MLP | $0.4074 \pm 0.0592$ |
| RNN (LFADS) | $0.5015 \pm 0.1085$ |
| Transformer (NDT) | $0.5272 \pm 0.0720$ |
| Transformer (PoYo) | $0.5944 \pm 0.0901$ |
| **RPNT** (from scratch) | $\mathbf{0.6358} \pm 0.0311$ |
| **RPNT** (pretrained) | $\mathbf{0.6612} \pm 0.0328$ |

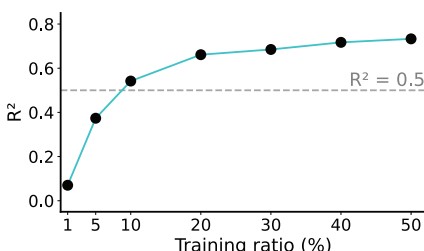

Figure 3: Decoding performance comparison across sites (B-CS) on the neuropixel dataset. The mean and standard deviation were calculated across all successful trials.

Figure 4: Sweep analysis for training splits.

We further investigated the generalization of RPNT for variations that include cross-site scenarios on the neuropixel dataset. This dataset is distinct from the public benchmark in recording modality (neuropixels vs. microelectrodes) and is particularly suitable for cross-site variation study since these recordings did not share neurons across sites (see appendix figure 2 for recording sites topology). Figure 3 summarizes the decoding performance of RPNT compared with existing baselines. RPNT (in FS-SFT regime) outperforms existing baselines at the downstream single-site (S17) velocity decoding task. Furthermore, pretraining RPNT improved performance (scratch: 0.6358 vs. pretrained: 0.6612). In addition, we swept train splits from $1\%$ to $50\%$ ($20\%$ by default), while keeping the test data to be the last $50\%$. We observe that even with $10\%$ of training split (half of the prior), pretrained RPNT is able to achieve reasonable performance (see Figure 4).

### 4.4 ABLATION STUDIES

| Method | T-RT | B-CS |
|---|---|---|
| Sinusoidal PE | 0.8260 | 0.6242 |
| RoPE | 0.8226 | 0.6484 |
| Learnable PE | 0.8305 | 0.6273 |
| **MRoPE** | **0.8515** | **0.6612** |

Table 2: Ablation study of position embeddings

| Method | T-RT | B-CS |
|---|---|---|
| Standard attention | 0.8024 | 0.5024 |
| Context-based attention | **0.8515** | **0.6612** |

Table 3: Ablation study of attention

| | Masking Strategy | | T-RT | B-CS |
|---|---|---|---|---|
| | Neuron ratio | Temporal ratio | | |
| | 0.25 | 0.25 | $0.8349 \pm 0.0948$ | $0.6549 \pm 0.0254$ |
| | 0.50 | 0.50 | $0.8437 \pm 0.0968$ | $0.6557 \pm 0.0316$ |
| Fixed | 0.75 | 0.75 | $0.8392 \pm 0.0975$ | $0.6593 \pm 0.0251$ |
| | 0.25 | 0.75 | $0.8414 \pm 0.0990$ | $0.6540 \pm 0.0279$ |
| | 0.75 | 0.25 | $0.8398 \pm 0.0922$ | $0.6594 \pm 0.0301$ |
| **Random** | $\mathcal{U}(0,1)$ | $\mathcal{U}(0,1)$ | $\mathbf{0.8515} \pm 0.1071$ | $\mathbf{0.6612} \pm 0.0328$ |

Table 4: Ablation study of masking strategies during pretraining.

We conducted ablation studies on T-RT task and the novel B-CS task in the FS-SFT regime to evaluate the contribution of novel components of position encoding, attention mechanism, and masking strategy to the performance of RPNT and report the results in Tables 2, 3 (see appendix for the mean $\pm$ std), and 4. Additional sweeping analyses, e.g., transformer layer, attention head, and context-based kernel size, are included in the appendix. **MRoPE.** We compared MRoPE against three alternative positional encoding approaches: sinusoidal PE, standard RoPE, and multi-dimensional

learnable PE. In Table 2, we compare MRoPE against alternative positional encodings and observe that MRoPE achieves the highest $R^2$ score. Improvement of $\approx 3\%$ over standard RoPE validates the extension of RoPE to multiple dimensions **Context-based Attention Mechanism.** Table 3 demonstrated the importance of context-dependent attention design. Replacing context-based attention with standard self-attention resulted in a substantial performance drop of $\approx 5\%$. This significant gap highlights the effectiveness of the adaptive kernel generation mechanism in handling the non-stationary nature of neural recordings (see appendix for kernel studies). **Random Uniform Masking Strategy.** Fixed masking ratios, whether symmetric or asymmetric across neuron and temporal dimensions, consistently underperformed the RPNT random uniform masking approach (Table 4).

## 5 DISCUSSION AND CONCLUSION

We present RPNT, a robust pretrained neural transformer comprised of three novel components (MRoPE, context-based attention mechanism, and uniform random masking strategy). Pretraining RPNT with SSL objectives eliminates the need for behavioral labels and allows for finetuning it in a few-shot regime (FS-SFT) in decoding downstream tasks (Gunel et al., 2020; Ding et al., 2023). Our experiments show that RPNT outperforms in generalization to a variety of downstream decoding tasks. These advances pose RPNT as a promising pathway toward neural foundation modeling for neural decoding and brain-computer interfaces. Our work also has limitations. First, our modeling was limited to cortical motor areas, excluding brain regions such as the frontal cortex and the basal ganglia, which play a key role in motor control (Trautmann et al., 2025). Future work should incorporate recordings from these regions to pretrain more comprehensive neural population models. Second, center-out and random reaching tasks represent simplified and stereotyped motor behaviors. Natural behaviors and cognition tend to be more complex across species and even include feedback interactions with the environment. Validating RPNT on such datasets will be the next step (Pei et al., 2021; Willett et al., 2023; Karpowicz et al., 2024; de Vries et al., 2023; Safaie et al., 2023; Banga et al., 2025). Last, RPNT was developed for a single modality. Developing multimodal RPNT will be the next work (Wang et al., 2024a; Zhang et al., 2025).

# 1 ETHICS STATEMENT

This work focuses on algorithm development for neural data. To evaluate our RPNT model, we used two datasets: a public dataset and a neuropixels dataset. The public dataset is from Perich et al. (2018). Data privacy and ethics protocols for experimental procedures related to neural data collection for this data were addressed in the original publication. The neuropixels dataset was collected at the University Institution and the National Primate Research Center. All procedures were conducted in compliance with the NIH Guide for the Care and Use of Laboratory Animals and were approved by the institutional Animal Care and Use Committee.

# 2 REPRODUCIBILITY STATEMENT

All hyperparameters, architectural details, and training configurations necessary for reproducing our results are provided in Appendix J. Data preprocessing steps and experimental protocols are described in Appendix I. Code will be made publicly available on GitHub upon paper acceptance.

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

APPENDICES

## A    THE USE OF LARGE LANGUAGE MODELS (LLMS)

We used the large language model GPT for checking grammar and spelling, and improving the clarity of our manuscript's language. All scientific content, experimental design, analysis, and interpretations are entirely the original work of the authors.

## B    BROADER IMPACT

Our results have a broad impact for future brain-computer interface (BCI) designs. Current BCI decoders typically process neural activity signals in isolation, missing the rich information encoded in historical recordings. Our RPNT's superior performance on held-out cases demonstrates that the pretraining strategy can substantially improve decoding robustness. Second, the computational visualization framework provides a data-driven tool to quantify and functional connectivity and directional information flow across brain sites. Third, while demonstrated on the motor cortex, our RPNT framework is applicable to broad BCI decodings with proper modification.

## C    FORMULATION OF MROPE

For simplicity, we use 3D-RoPE as a concrete example of the general MRoPE illustration. The 4D-RoPE can be found in section C.4. For neural recordings from $S$ sites over $T$ timesteps, we require positional encodings $\mathbf{PE}(s, t, d) \in \mathbb{R}^{d_{model}}$ where $d$ is the model embedding dimension. A good positional encoding is preferred to satisfy two key properties: (1) *site specific*: $\mathbf{PE}(s_i, t, d) \neq \mathbf{PE}(s_j, t, d)$ for $i \neq j$, ensuring distinct representations for different site locations; (2) *zero-shot generalization*: the encoding function can generate reasonable postional embedding giving any new sites $\mathbf{s}_{new} = (x_{new}, y_{new})$ (or configurations in MRoPE case).

### C.1    EXTENSION FROM STANDARD ROPE TO 3D-ROPE

The key insight of the standard RoPE is that attention scores depend only on relative distances. Thus, combining the RoPE and session configuration (spatial coordinates $(x, y)$ for site $s$ and temporal position $t$), our goal is to build 3D-RoPE, in which the attention score can depend on relative spatial and temporal distances: $f(|x_i - x_j|, |y_i - y_j|, |t_i - t_j|)$. For position $(x, y, t)$, the complete 3D-RoPE transformation is a block-diagonal matrix:

$$\mathbf{R}_{3D}^d = \begin{bmatrix} \mathbf{R}_x(x) & \mathbf{0} & \mathbf{0} \\ \mathbf{0} & \mathbf{R}_y(y) & \mathbf{0} \\ \mathbf{0} & \mathbf{0} & \mathbf{R}_t(t) \end{bmatrix} \in \mathbb{R}^{d \times d} \tag{1}$$

where each sub-matrix $\mathbf{R}_{m \in \{x,y,t\}}$ consists of 2×2 rotation blocks:

$$\mathbf{R}_m(m) = \begin{bmatrix} \cos(m\theta_0^{(m)}) & -\sin(m\theta_0^{(m)}) & & & \\ \sin(m\theta_0^{(m)}) & \cos(m\theta_0^{(m)}) & & & \\ & & \ddots & & \\ & & & \cos(m\theta_{\frac{d}{6}-1}^{(m)}) & -\sin(m\theta_{\frac{d}{6}-1}^{(m)}) \\ & & & \sin(m\theta_{\frac{d}{6}-1}^{(m)}) & \cos(m\theta_{\frac{d}{6}-1}^{(m)}) \end{bmatrix} \tag{2}$$

### C.2    RELATIVE POSITION PROPERTY PROOF

3D-RoPE maintains the RoPE property that the attention scores depend only on relative configuration. For two configureations $(x_i, y_i, t_i)$ and $(x_j, y_j, t_j)$:

$$\mathbf{q}_i^T \mathbf{k}_j = (\mathbf{R}_{3D}(x_i, y_i, t_i)\mathbf{q})^T (\mathbf{R}_{3D}(x_j, y_j, t_j)\mathbf{k}) \tag{3}$$

$$= \mathbf{q}^T \mathbf{R}_{3D}^T(x_i, y_i, t_i)\mathbf{R}_{3D}(x_j, y_j, t_j)\mathbf{k} \tag{4}$$

$$= \mathbf{q}^T \mathbf{R}_{3D}(x_j - x_i, y_j - y_i, t_j - t_i)\mathbf{k} \tag{5}$$

This follows from the rotation composition property: $\mathbf{R}^T(\alpha)\mathbf{R}(\beta) = \mathbf{R}(\beta-\alpha)$. This design ensures **site-specificity** since different spatial coordinates produce different rotations, and enables **zero-shot generalization** as the rotation operations naturally handle arbitrary continuous spatial coordinates.

### C.3 OTHER POSITIONAL EMBEDDING BASELINES

**Original RoPE:**

$$
\mathbf{R}_{\Theta,m}^d = \begin{bmatrix}
\cos m\theta_0 & -\sin m\theta_0 & 0 & 0 & \cdots & 0 & 0 \\
\sin m\theta_0 & \cos m\theta_0 & 0 & 0 & \cdots & 0 & 0 \\
0 & 0 & \cos m\theta_1 & -\sin m\theta_1 & \cdots & 0 & 0 \\
0 & 0 & \sin m\theta_1 & \cos m\theta_1 & \cdots & 0 & 0 \\
\vdots & \vdots & \vdots & \vdots & \ddots & \vdots & \vdots \\
0 & 0 & 0 & 0 & \cdots & \cos m\theta_{d/2-1} & -\sin m\theta_{d/2-1} \\
0 & 0 & 0 & 0 & \cdots & \sin m\theta_{d/2-1} & \cos m\theta_{d/2-1}
\end{bmatrix}
\tag{6}
$$

**Learnable PE** We also implement an improved version of the traditional learnable PE (Devlin et al., 2019) so that it is 1) learnable; 2) has a simple mechanism for the configuration generalization. Thus, we employ a learned MLP that directly maps spatial-temporal coordinates to the embedding space:

$$
\mathbf{PE}(s, t, d) = \text{MLP}_{pos}([x \cdot \alpha, y \cdot \alpha, t])
\tag{7}
$$

where $\alpha$ is a spatial scaling factor. In our experiments, we set the $\alpha = 1$ for simplicity.

**Sinusoidal PE** Last, we also incorporate the standard sinusoidal positional embedding (Vaswani et al., 2017) in our ablation study.

### C.4 4D-RoPE IN THE PUBLIC BENCHMARK

We showed the following rotation matrix for $M = 4D$ that was used on the public benchmark

$$
\mathbf{R}_{4D}^d = \begin{bmatrix}
\mathbf{R}_{\text{task}}(x) & \mathbf{0} & \mathbf{0} & \mathbf{0} \\
\mathbf{0} & \mathbf{R}_{\text{subject}}(y) & \mathbf{0} & \mathbf{0} \\
\mathbf{0} & \mathbf{0} & \mathbf{R}_{\text{recording time}}(t) & \mathbf{0} \\
\mathbf{0} & \mathbf{0} & \mathbf{0} & \mathbf{R}_t(t)
\end{bmatrix} \in \mathbb{R}^{d \times d}
\tag{8}
$$

where each entry was constructed similarly to equation 2. Different from the Neuropixel datasets, in which the $(x, y)$ coordinates have values, the public benchmark was string-based meta information. Therefore, we used the discrete embeddings for representation. Specifically, we used i) $[0, 1]$ to embed behavior types $\{\text{CO}, \text{RT}\}$; ii) $[0, 1, 2, 3]$ to embed subject $\{c, j, m, t\}$; iii) $[0, 1]$ range normalization to the recording time (day/month/year). We further envision that using learnable MLPs for the string-based meta information may further improve the embedding representation for MRoPE. However, it is more about the incremental improvement of our MRoPE, and thus will not be investigated in this paper.

## D LIGHTWEIGHT DECODER

Our main goal in RPNT is to get the pretrained transformer encoder, which later can be either fine-tuned for downstream tasks or freezed for data-driven functional connectivity visualization. The decoder is thus designed to be lightweight $\text{Decoder}_\theta : \mathbb{R}^{B \times T \times D} \to \mathbb{R}^{B \times T \times N}$ maps encoded representations to Poisson rate parameters through two feedforward blocks and an output projection:

$$
\mathbf{Z} = \text{FFN}_2(\text{FFN}_1(\text{LayerNorm}(\mathbf{z})))
\tag{9}
$$

$$
\boldsymbol{\lambda} = \text{Softplus}(\text{Linear}(\mathbf{Z}_2))
\tag{10}
$$

where each block is defined as:

$$
\text{FFN}_1(\mathbf{x}) = \text{Dropout}(\text{GELU}(\mathbf{x}\mathbf{W}_1 + \mathbf{b}_1)), \quad \mathbf{W}_1 \in \mathbb{R}^{D \times \frac{D}{2}}
\tag{11}
$$

$$
\text{FFN}_2(\mathbf{x}) = \text{Dropout}(\text{GELU}(\mathbf{x}\mathbf{W}_2 + \mathbf{b}_2)), \quad \mathbf{W}_2 \in \mathbb{R}^{\frac{D}{2} \times \frac{D}{4}}
\tag{12}
$$

$$
\text{Linear}(\mathbf{x}) = \mathbf{x}\mathbf{W}_{\text{out}} + \mathbf{b}_{\text{out}}, \quad \mathbf{W}_{\text{out}} \in \mathbb{R}^{\frac{D}{4} \times N}
\tag{13}
$$

| Point | Masked Autoencoding | Causal Masked Neural Modeling |
|---|---|---|
| Information flow | Bidirectional | Unidirectional |
| Goal | Interpolation | Extrapolation |
| Application | Offline (Denoising) | Online (Prediction) |

Table 1: Comparison between MAE and our proposed causal masked neural modeling

| Method | Cross-Site (B-CS) |
|---|---|
| w.o contrastive loss | $0.6267 \pm 0.0277$ |
| **w. contrastive loss** | **0.6612** $\pm 0.0328$ |

Table 2: Ablation study on contrastive loss

The softplus activation ensures positive rate parameters suitable for Poisson likelihood modeling of neural spiking activity. Although this MLP-based decoder is lightweight compared to the sophisticated transformer decoder design in (Vaswani et al., 2017), it enables efficient training while maintaining decoding capacity.

# E  PRETRAINING DETAILS

## E.1  COMPARISON WITH THE STANDARD MAE AND OUR OBJECTIVE

We compare our proposed causal masked neural modeling with the prior MAE approach in table 1.

## E.2  CROSS-SITE CONTRASTIVE LEARNING DETAILS

For each site representation $\mathbf{z}_{s_i,t}$ from site $s_i$ at time $t$, we define the Positive instances and Negative instances as the same-site neural representation and the different sites neural representation. For simplicity, we first average the representation for all time $t$. The temperature $\tau$ controls the smoothness of the similarity distributions; we set $\tau = 0.1$ consistently across all experiments on the private neuralpixel dataset. Thus, we contrast representations from different sites:

$$\mathcal{L}_{\text{contrast}} = -\frac{1}{S} \sum_{i=1}^{S} \log \frac{\exp(\text{sim}(\bar{\mathbf{z}}_i, \bar{\mathbf{z}}_i)/\tau)}{\sum_{j=1}^{S} \exp(\text{sim}(\bar{\mathbf{z}}_i, \bar{\mathbf{z}}_j)/\tau)} \tag{14}$$

We showed the ablation study for the contrastive loss in table 2. Our results showed that by contrastive loss, it encourages the RPNT model to learn more robust neural representations that have better downstream decoding performance.

# F  COMPARE WITH BASELINE MODELS

We compared our RPNT with a few popular decoding models as baselines. Specifically, we included the Wiener filter (Van Drongelen, 2018) and several standard machine learning models such as MLP (Glaser et al., 2020), GRU (Cho et al., 2014), S4D (Gu et al., 2021), Mamba (Gu & Dao, 2023), transformer (Vaswani et al., 2017; Devlin et al., 2019) that were trained from scratch on the downsrteam single session. We also included two baseline pretrained models: NDT2 (Ye et al., 2023) and the latest POSSM (Ryoo et al., 2025). Further, we acknowledge the author's contribution (Ryoo et al., 2025) for implementing and testing all the baseline models in Table 1.

# G  VISUALIZATION

We examined the attention-based FC map $\mathbf{C} \in \mathbb{R}^{T \times S \times S}$ from the spatial encoder for the Neuropixel dataset. Appendix figure 1 showed the time evolution (0ms, 300ms, 600ms) of FC during the reaching period. We observed that site 6 remained quite active, whereas large site numbers, e.g., S10 and later, seemed inactive. Further, it revealed a small network interaction within the large PMd-M1

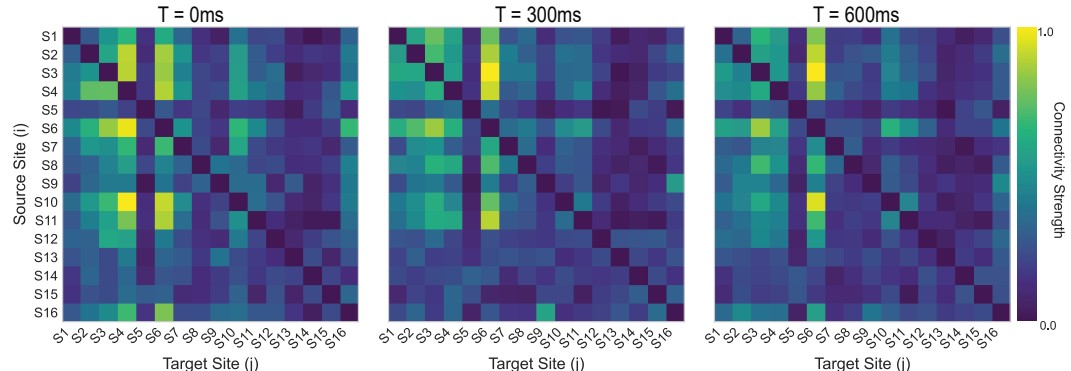

Figure 1: Spatial attention map provided data-driven brain network insights for motor behaviors.

| Transformer layers | Cross-Task (T-RT) |
|---|---|
| 2 | $0.8482 \pm 0.0955$ |
| 3 | $0.8432 \pm 0.1004$ |
| 4 | $0.8515 \pm 0.1071$ |
| 5 | $0.8431 \pm 0.0914$ |
| 6 | $0.8461 \pm 0.0942$ |

Table 3: Sweeping study on transformer layer

| Attention heads | Cross-Task (T-RT) |
|---|---|
| 4 | $0.8320 \pm 0.1054$ |
| 8 | $0.8531 \pm 0.0931$ |
| 16 | $0.8515 \pm 0.1071$ |
| 32 | $0.8509 \pm 0.1027$ |
| 64 | $0.8505 \pm 0.0935$ |

Table 4: Sweeping study on attention head

| Kernel size | Cross-Task (T-RT) |
|---|---|
| $[3, 3]$ | $0.8317 \pm 0.0945$ |
| $[7, 7]$ | $0.8481 \pm 0.0954$ |
| $[9, 9]$ | $0.8515 \pm 0.1071$ |
| $[11, 11]$ | $0.8411 \pm 0.0966$ |
| $[15, 15]$ | $0.8479 \pm 0.0900$ |

Table 5: Sweeping study on kernel size

region, which provided data-driven insights into neural mechanisms of motor behavior. These results may provide insight for future intervention investigation of brain sites to verify the data-driven observations.

## H SWEEPING ANALYSIS FOR RPNT MODEL

We showed the sweeping analysis result for our RPNT model on the T-RT task, including temporal transformer layers (see table 3), attention heads (see table 4), and kernel sizes in context-based attention (see table 5). Further, the standard deviation included ablation studies tables were listed (see table 6 and table 7).

| Method | Cross-Task (T-RT) | Cross-Site (B-CS) |
|---|---|---|
| Sinusoidal PE | $0.8260 \pm 0.0894$ | $0.6242 \pm 0.0267$ |
| RoPE | $0.8226 \pm 0.0941$ | $0.6484 \pm 0.0074$ |
| Learnable PE | $0.8305 \pm 0.0917$ | $0.6273 \pm 0.0398$ |
| **MRoPE** | $\mathbf{0.8515} \pm 0.1071$ | $\mathbf{0.6612} \pm 0.0328$ |

Table 6: Ablation study of positional encoding methods.

| Method | Cross-Task (T-RT) | Cross-Site (B-CS) |
|---|---|---|
| Standard attention | $0.8024 \pm 0.0896$ | $0.5024 \pm 0.0388$ |
| **Context-based attention** | $\mathbf{0.8515} \pm 0.1071$ | $\mathbf{0.6612} \pm 0.0328$ |

Table 7: Ablation study of attention mechanisms.

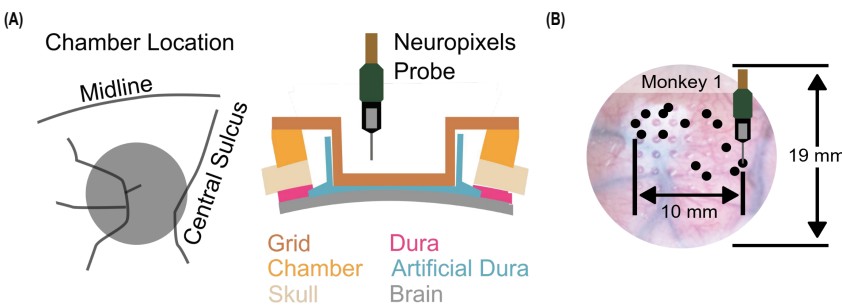

Figure 2: Neuropixel data collection setup. (A) Chamber location and neuropixel probe. (B) Neuropixels probe insertion locations (black dots) on the cortical surface. Each black dot stands for a different recording site.

# I    NEUROPIXEL DATA COLLECTION AND PREPROCESSING

## I.1    NEURALPIXEL DATA COLLECTION

We utilized high-density Neuropixel recordings (see appendix figure 2 for neuropixel setup and appendix table 8 for each recording information) from a NHP performing center-out reaching tasks. The dataset is comprised of 17 recordings at different sites distributed across the PMd and M1 regions across experiment sessions spanning approximately one year.

## I.2    PREPROCESSING PIPELINE

We required batch-like neural data for transformer model pretraining. However, the raw data across heterogeneous recording sites have varying trial counts, neuron populations, and recording times. Let $\mathcal{S} = \{S_1, S_2, \ldots, S_K\}$ denote a collection of $K$ recording sites, where each site $S_i$ contain neural spike data $\mathbf{X}_i \in \mathbb{R}^{C_i \times T_i \times N_i}$ with $C_i$ counts of successful trials, $t_j$ time points, and $N_i$ recording neurons after spike sorting. Our objective was to construct a unified dataset $\mathcal{D} = \{(\mathbf{X}_{k=1}^K), \mathbf{L}\}$ where $\mathbf{X}^{(k)} \in \mathbb{R}^{C \times T \times N}$ represented standardized neural activity for each site and $\mathbf{L} \in \mathbb{R}^{L \times 2}$ encode spatial location (coordinates).

Therefore, we implemented a systematic approach to load multi-site neuropixel recordings. Given a metadata file containing site locations $\mathbf{L}_i = (x_i, y_i) \in \mathbb{R}^2$ and corresponding data files, we filter sites based on data availability and quality criteria. Each site's spike data is loaded as $\mathbf{X}_i \in \mathbb{R}^{C_i \times T_i \times N_i}$. We first extract only the 1s reaching period data and apply it with $20ms$ bin size. This process gives us uniform temporal dimension ($T = 50$). Next, we standardize neuron numbers across heterogeneous sites while creating data variety through neuron subset sampling. For each train/valid/test split $s$ (based on trials) and site $i$ with shape $\mathbf{X}_i^{(s)} \in \mathbb{R}^{C_i^{(s)} \times T \times N_i}$, we apply the following pipeline:

---

**Algorithm 1** Data Preprocessing Pipeline on the Neuropixel Dataset

---

**Require:** Site data $\mathbf{X}_i \in \mathbb{R}^{C_i \times T \times N_i}$, target neurons $N$, sample times $M$, target trial counts $C_{\text{target}}^{(s)}$

**Ensure:** Standardized data $\hat{\mathbf{X}}_i^{(s)} \in \mathbb{R}^{T_{\text{target}}^{(s)} \times T \times N}$

1: **Step 1: Split** - Apply train/val/test split (e.g., $80\%/10\%/10\%$) to get $\mathbf{X}_i^{(s)}$
2: **Step 2: Neuron Multi-Sampling**
3: $N_{\text{total}} \leftarrow N \cdot M$
4: **if** $N_i \geq N_{\text{total}}$ **then**
5: $\quad \boldsymbol{\nu} \leftarrow \text{sample\_without\_replacement}(N_i, N_{\text{total}})$
6: **else**
7: $\quad \boldsymbol{\nu} \leftarrow \text{sample\_with\_replacement}(N_i, N_{\text{total}})$
8: **end if**
9: $\mathbf{Y} \leftarrow \mathbf{X}_i^{(s)}[:, :, \boldsymbol{\nu}]$ {Neuron sampling}
10: $\mathbf{Z} \leftarrow \text{reshape}(\mathbf{Y}, (C_i^{(s)} \cdot M, T, N))$ {Multi-sampling}
11: **Step 3: Trial Sampling for Target Matching**
12: $T_{\text{available}} \leftarrow T_i^{(s)} \cdot M$
13: **if** $T_{\text{available}} \geq T_{\text{target}}^{(s)}$ **then**
14: $\quad \boldsymbol{\tau} \leftarrow \text{sample\_without\_replacement}(C_{\text{available}}, C_{\text{target}}^{(s)})$
15: **else**
16: $\quad \boldsymbol{\tau} \leftarrow \text{sample\_with\_replacement}(C_{\text{available}}, C_{\text{target}}^{(s)})$
17: **end if**
18: $\hat{\mathbf{X}}_i^{(s)} \leftarrow \mathbf{Z}[\boldsymbol{\tau}, :, :]$ {Final trial sampling}

---

After neuron multi-sampling, we apply split-specific trial sampling to match target trial counts:

$$C_{\text{target}}^{(s)} = \begin{cases} C_{\text{train}} & \text{if } s = \text{train} \\ C_{\text{min}} & \text{if } s \in \{\text{val}, \text{test}\} \end{cases} \tag{15}$$

where $T_{\text{train}}$ is the target training sample count and $T_{\text{min}}$ is the minimal validation/test count to prevent oversampling. Last, we aggregate standardized site data into consistent dimension tensors

$$\mathbf{X}_{\text{sites}}^{(s)} = \text{stack}([\hat{\mathbf{X}}_1^{(s)}, \hat{\mathbf{X}}_2^{(s)}, \dots, \hat{\mathbf{X}}_K^{(s)}]) \in \mathbb{R}^{K \times C_{\text{target}}^{(s)} \times T \times N} \tag{16}$$

$$\mathbf{X}^{(s)} = \text{transpose}(\mathbf{X}_{\text{sites}}^{(s)}, (1, 0, 2, 3)) \in \mathbb{R}^{C_{\text{target}}^{(s)} \times K \times T \times N} \tag{17}$$

Our data processing approach ensures strict separation ($\mathbf{X}_i^{\text{train}} \cap \mathbf{X}_i^{\text{val}} \cap \mathbf{X}_i^{\text{test}} = \emptyset, \quad \forall i \in \{1, \dots, K\}$) while tensors maintain consistent dimensionality, i.e., $\mathbf{X}^{(s)} \in \mathbb{R}^{B^{(s)} \times K \times C \times N}, \quad \forall s \in \{\text{train}, \text{val}, \text{test}\}$, where $B^{(s)} = \tilde{C}^{(s)}$ represents the batch dimension for split $s$.

Raw trial counts range from 200 to 1417 per site; raw neuron counts range from 83 to 703 per site. Our preprocessing pipeline standardizes all site data, which generates 80,000 (16*5000) training samples for our RPNT pretraining and efficient validation/test data. Further, our pipeline generates the dataset being agnostic of neuron numbers and neuron orders; rather, it only depends on the target neurons $N$ that one can specify. Therefore, our model trained on this dataset can handle arbitrary numbers of neurons via the sampling strategy in our pipeline. To this end, table 8 and table 9 summarized the pretraining data statistics before and after the preprocessing.

## J  HYPERPARAMETERS AND COMPUTATIONAL RESOURCES

We showed the detailed hyperparameter setup for the RPNT model pretraining and SFT in table 10 and table 11, respectively. Pretraining of RPNT was trained using A40 GPUs, consuming 3GB and 9GB of GPU memory for the public benchmark and neuropixel dataset, respectively, which takes around 12 hours for training. We used the best checkpoint based on the early stopping criteria for the downstream SFT. For the benchmark, it took 1 hour for C-CO and 30 minutes for T-CO and T-RT. For the neuropixel data, it took 15 mins. Our SFT results are reported based on the last epoch.

| Site ID | Coordinates | Trials | Time bins | Neurons | Site labels |
|---|---|---|---|---|---|
| 9940 | (-1, 1) | 1286 | 50 | 328 | S13 |
| 10802 | (3, 2) | 1200 | 50 | 703 | S11 |
| 10812 | (-3, 2) | 1200 | 50 | 511 | S9 |
| 10820 | (-2, 4) | 1086 | 50 | 394 | S3 |
| 10828 | (-1, 4) | 1417 | 50 | 329 | S4 |
| 12269 | (3, -4) | 752 | 50 | 252 | S16 |
| 12290 | (5, 2) | 763 | 50 | 367 | S12 |
| 13153 | (1, 5) | 415 | 50 | 275 | S2 |
| 13122 | (2, 4) | 674 | 50 | 83 | S6 |
| 13239 | (-1, 5) | 967 | 50 | 154 | S1 |
| 13256 | (2, -4) | 1024 | 50 | 255 | S15 |
| 13272 | (-1, 3) | 1056 | 50 | 187 | S7 |
| 14116 (Test) | (3, -2) | 789 | 50 | 258 | S17 |
| 14139 | (2, 2) | 865 | 50 | 125 | S10 |
| 14824 | (2, 3) | 200 | 50 | 354 | S8 |
| 14878 | (3, -3) | 200 | 50 | 309 | S14 |
| 14891 | (0, 4) | 200 | 50 | 106 | S5 |

Table 8: Site-by-site breakdown showing data characteristics

| Split | Samples | Shape | Sampling Strategy |
|---|---|---|---|
| Training | 80,000 | $(5000, 16, 50, 50)$ | Full sampling |
| Validation | 3,200 | $(200, 16, 50, 50)$ | Minimal sampling |
| Test | 3,200 | $(200, 16, 50, 50)$ | Minimal sampling |

Table 9: Dataset split characteristics after preprocessing. Format: (Trials, Sites, Time, Neurons).

| Parameter | microelectrode benchmark | Neuropixel dataset |
|---|---|---|
| Model dimension | 512 | 384 |
| Temporal layers | 4 | 4 |
| Spatial layers | N/A | 2 |
| Attention heads | 16 | 12 |
| Kernel size | $[9, 9]$ | $[9, 9]$ |
| Dropout rate | 0.1 | 0.1 |
| Batch size | 64 | 32 |
| Epoch | 100 | 100 |
| Warm-up epoch | 50 | 10 |
| Weight decay | 0.01 | 0.01 |
| Learning rate | $5 \times 10^{-5}$ | $5 \times 10^{-5}$ |
| Gradient clip | 1.0 | 1.0 |
| Contrastive loss ($\lambda$) | N/A | 0.1 |
| Random seed | 3407 | 3407 |

Table 10: Hyperparameter setup for pretraining RPNT on benchmark and neuropixel dataset

| Parameter | microelectrode benchmark | Neuropixel dataset |
|---|---|---|
| Dropout rate | 0.1 | 0.1 |
| Batch size | 32 | 32 |
| Epoch | 200 | 200 |
| Weight decay | 0.01 | 0.01 |
| Learning rate | $1 \times 10^{-4}$ | $1 \times 10^{-4}$ |
| Gradient clip | 1.0 | 1.0 |
| Random seed | 3407 | 3407 |

Table 11: Hyperparameter setup for SFT of RPNT on both datasets