# OpenReview forum: "RPNT: Robust Pre-trained Neural Transformer - A Pathway for Generalized Motor Decoding"
_ICLR.cc/2026/Conference — Submitted to ICLR 2026_

### Official Review · Reviewer_c4qq · 2025-10-30

**Soundness:** 2
**Presentation:** 2
**Contribution:** 2
**Rating:** 4
**Confidence:** 3

**Summary:**

To improve motor decoding from neural activity, this paper introduced several new modifications to the standard transformer architecture. Multidimensional Rotary Positional Embedding (MRoPE) allows the model to embed more recording meta data. Context-based attention mechanism uses convolutional kernels on attention maps to capture local temporal structures and handle neural non-stationarity. Uniform random masking across time and neuron dimensions that avoids fixed masking hyperparameters and improves representation learning. RPNT was pretrained on 2 datasets and tested across cross-session, cross-task, cross-subject, and cross-site generalization tasks. It consistently outperformed existing decoding models (e.g., POYO, POSSM).

**Strengths:**

1. MRoPE and context-based attention are domain-specific modification of the transformer design that directly address neural recording variability and non-stationarity.
2. The experiments cover multiple generalization settings (cross-session, cross-subject, cross-task, cross-site).
3. This paper shows practical utility in low-data regimes, which is relevant for BCIs.

**Weaknesses:**

1. The authors claimed that this work is a step toward neurofoundation models, but the amount of pretraining data used is relatively limited. Many of the model’s design choices seem motivated by BCI-specific applications, which may limit its applicability to other datasets. A truly promising foundation model should show scalability beyond architectural innovations, yet the paper does not thoroughly explore the scaling behavior of RPNT.

2. The model diagram could be improved for clarity. As it stands, it is difficult to fully understand the model’s components and data flow based only on the figure and its caption.

3. The study focuses exclusively on monkey motor tasks, which restricts its generality. It would be good to test RPNT on other species and task domains, such as the IBL or Allen decision-making datasets. This would test whether the model generalizes beyond motor cortex data, especially given that monkey and mouse neural recordings can have large differences.

**Questions:**

1. The paper proposes a uniform random masking strategy, which outperforms fixed-ratio masking. I wonder whether this contributes significantly to the decoding performance, since masking can help prevent overfitting (especially when the single- and multi-session datasets are not very large). Could the authors test a similar masking strategy in POYO, which currently uses fixed-ratio coordinated dropout, to see if it improves POYO's decoding performance?

2. The authors state that the local convolutional attention helps address non-stationarity in neural data, but no specific experiments directly test this claim. Could the authors include a cross-day analysis to better motivate this design choice, beyond the existing cross-session evaluations?

---

> ### Author Response · Authors · 2025-11-28
> **We appreciate the reviewer’s thoughtful review, feedback and support for our work. We provide  a point-to-point response to each comment below and we are hopeful that these resolve reviewer concerns. We are happy to further clarify in case there are additional questions.**
>
> **W1**
>
> In the discussion section, we address the future pathway of RPNT toward the neurofoundation model: “These advances pose RPNT as a promising pathway toward neural foundation modeling for neural decoding and brain-computer interfaces”. This sentence describes the potential implications of our work toward designing future neural foundation models. Beyond such a forward-looking goal, while RPNT is showing promise toward a neurofoundation model, this is not the main focus of our work, and we have been conservative in classifying RPNT as such a model. Our main focus remains on the idea of pretraining and fine-tuning frameworks and aims to achieve generalization of motor decoding.
>
> **W2**
>
> We thank the reviewer for the comment. We added an explanation in the model diagram figure caption as below:.
> MRoPE incorporates experimental metadata into the transformer positional embeddings. The context-based attention captures local temporal structures. The input spike activity is randomly masked according to a uniform distribution for causal masked neural modeling. The masked input spike activity is projected into neural embeddings, forming Q, K, and V representations. MRoPe is applied to get the rotary queries and keys. Attention scores are modulated by context-based convolutional kernels and combined with causal masking before aggregating with values.
>
> **W3**
>
> We will incorporate discussion of the scope and focus of our experiments into the discussion and future work, as below. Validation of our RPNT approach on different species and cognitive tasks is not within the scope of the current work but will be an interesting future direction.
> Natural behaviors and cognition tend to be more complex across species and even include feedback interactions with the environment. Exploration of  RPNT generalization and performance on such datasets is one of the plausible future directions for this work  [1,2,3,4,5].
>
> [1] Pei, Felix, et al. "Neural latents benchmark'21: evaluating latent variable models of neural population activity." arXiv preprint arXiv:2109.04463 (2021).
>
> [2] Karpowicz, Brianna M., et al. "Few-shot algorithms for consistent neural decoding (falcon) benchmark." Advances in Neural Information Processing Systems 37 (2024): 76578-76615.
>
> [3] de Vries, Saskia EJ, Joshua H. Siegle, and Christof Koch. "Sharing neurophysiology data from the Allen Brain Observatory." Elife 12 (2023): e85550.
>
> [4] Safaie, Mostafa, et al. "Preserved neural dynamics across animals performing similar behaviour." Nature 623.7988 (2023): 765-771.
>
> [5] Banga, Kush, et al. "Reproducibility of in vivo electrophysiological measurements in mice." Elife 13 (2025): RP100840.
>
> **Q1**
>
> We thank the reviewer for raising an interesting question of whether the uniform random masking strategy that we propose can also be applicable to the current PoYo and PoSSM models. We ran a feasible test of random masking strategy integrated into PoYo and tested it on the downstream neuropixel dataset. Our results confirm that a random masking strategy could be effective for the PoYo model, as we show below. We would like to note that this is a partial test, since to show masking effectiveness, it requires initializing a new PoYo/PSSSM pretraining, which takes a large amount of computational resources and requires more datasets beyond the time of the rebuttal.
> | Method | B-CS |
> |--------|------|
> | PoYo with random uniform masking | 0.6331 ± 0.0940 |
> | PoYo with fixed masking (p~0.5) | 0.5944 ± 0.0901 |
>
> **Q2**
>
> In section 4.4 in the ablation study and Table 3, we show that with context-based attention, RPNT with context-based attention s performs better than the standard attention. This implies that the context-based kernel handles the underlying nonstationarity of brain activity. T-RT tasks include 6 recording sessions from different days, such that the cross-session task in this case also includes cross-day variation.

---

### Official Review · Reviewer_sU3Q · 2025-10-30

**Soundness:** 1
**Presentation:** 1
**Contribution:** 2
**Rating:** 2
**Confidence:** 5

**Summary:**

This paper proposes a pre-trained neural transformer for motor decoding called RPNT, combing multidimensional rotary embedding (RoPE) to incorporate electrode configurations, context-based attention for non-stationality and SSL objectives based on masked autoencoding and contrastive learning. However, there are key details missing regarding both the model components and experimental setup, which are essential for comprehensive understanding and assessment. Hence I give a rating of non-acceptance.

**Strengths:**

- Proposed multidimensional RoPE and context-based attention are novel and valuable additions
- Evaluations on two datasets show competitive motor decoding performance
- Ablation studies on the proposed components justifying their importance

**Weaknesses:**

1. The main result table of the first monkey dataset (Table 1) has all baseline results taken from two previous papers, but based on my understanding the results of RPNT and baselines are not using the same split nor evaluation procedure and nor finetuning strategy, and at least one of the papers was not using the brainset data split that the authors used in this paper. Overall, there may be unfair comparisons.
2. Detailed description on key components of RPNT is lacking, and this makes it difficult to properly assess and compare:
    - The model takes trial-aligned inputs, but the first monkey dataset has variable number of neurons in each session
    - Context-based attention needs historical data but it is never officially defined
    - Bin size used in the model is not mentioned, making it impossible to judge whether all available outputs are included in the evaluation
    - The model is mentioned as a causal one but the introduction of conv2d layer on attention matrix makes the overall operation non-causal.
    - Subject similarity in multidimensional RoPE is hand-picked, introducing inductive bias that the evaluation subject is similar to and drawing insights from one of the subjects in the training data

**Questions:**

Regarding weakness number 1:

1. After a quick check on cited publications, it seems the baseline scores in Table 1 are taken from cited publications POYO & POSSM, where one baseline Wiener Filter is taken from POYO paper and the rest 13 baselines are taken from POSSM paper. Could the authors clarify on this point and if so please provide proper citation to the original paper. Please see the links as below
   - POSSM's Table 1 from: https://arxiv.org/abs/2506.05320v1
   - POYO's Table 2 from: https://proceedings.neurips.cc/paper_files/paper/2023/file/8ca113d122584f12a6727341aaf58887-Paper-Conference.pdf
2. I have experience on the said first monkey datasets, and based on my understanding, the experimental setups in this paper, POYO and POSSM are not the same. As mentioned in the POSSM paper, they used a special causal evaluation to report the results and their splits were not shuffled (as shown in Figure 5 of POSSM paper), and this were supposedly different from the POYO paper, because they both reported POYO-1 results but with different values. In addition, _brainset_ split used by this paper is shuffled (as can be seen in the official implementation), further making it different from the data used in POSSM. Hence I don’t believe it is a fair comparison in Table 1 as RPNT and baselines are likely training and evaluating on different data. Could the authors clarify on what is the exact experimental setup used to run RPNT?
3. In section 4.2 it is claimed that all baselines were using Full-SFT, meaning all available training sessions were used for each finetuning result. I believe that this is incorrect, and only single session data was used for those baselines’ finetuning results.
4. Could authors comment on how the standard deviations were calculated in both tables (marked in the paper as Table 1 and Figure 3)?
5. To my knowledge there are no currently available benchmarking tools for the first monkey dataset, therefore it is very hard to determine whether two results are indeed a fair comparison. I believe it is needed to provide ample evidence that the same setups are being used among multiple papers, including detailed text description and / or code examples, if one want to include the published results.

Regarding weakness number 2:

6. As shown in line 240 page 5, RPNT expects inputs with a fixed number of neurons (as in D), but the first monkey dataset is not trial-aligned, and their individual sessions have different numbers of neurons. Could the author elaborate the preprocessing steps on this dataset to maintain the same of neurons?
7. As mentioned in section 2, the context-based attention takes historical data as input and outputs convolution kernels. This is an important component as indicated in the ablation study, but the historical data is not properly defined thoughout the paper. Please clarify on how to construct the historical data and what are T and T_hist?
8. Please comment on what are the bin sizes used in the study. RPNT gives one output per bin (mentioned in section 3.5), while in baselines results (POYO & POSSM) all outputs were evaluated. Could authors comment on how to ensure that all outputs are being evaluated in RPNT? Were any aggregation techniques (like averaging) used on the outputs?
9. It is claimed in the paper that RPNT is doing causal prediction, because of causal masking. However I believe that the introduction of conv2D on attention matrix violates causality constraint, in that the current kv value is affected by both earlier and later kv value. Please correct me if I have misunderstanding. I think it is probably more faithful to the causal operation if switching the order between causal masking and convolution in equation (5). Have the authors tried this variant?
10. For the multidimensional RoPE, as shown in appendix C.4, for the first monkey dataset, 0, 1, 2, 3 are assigned to monkey c, j, m, t, respectively. Because RoPE is a relative positional embedding, this formulation provides an inductive bias that monkey t is more alike to monkey m, compared to monkey c. Could the author comment on why it was chosen in this way? In addition, when more diverse datasets are included in the training, this doesn't seem to be a scalable way, as it needs to hand-pick the relationship between various tasks and various subjects.
11. Was contrastive learning objective only used in the neuropixel dataset? Since it is an important component in loss function, I believe it would be better to describe it in the main text.

Other questions:

12. Since both datasets are monkeys doing similar motor tasks with recording on similar regions, it would be interesting to see how will the model perform when transferring from one to the other?
13. In neuropixel S17, the train, valid, test splits in use were 0.2, 0.3 and 0.5, could the authors provide insights on why the specific splits? Also it would be interesting to see how the baselines would perform with up to 50% of the data, compared to the proposed method.
14. Please fix the typo in Table 1 (T-RT std of o-POSSM-Mamba & o-POSSM-GRU).
15. The table with neuropixel results is wrongly marked as Figure 3, please correct it.
16. Please define acronym when it is first used, i.e. line 319 page 6.

---

> ### Author Response · Authors · 2025-11-28
> **We thank the reviewer for their thoughtful review and feedback. We provide  a point-to-point response to each comment below and we are hopeful that these resolve reviewer concerns.  We are happy to further clarify in case there are additional questions.**
>
> **W1: Q1 + Q2**
>
> We appreciate the comment by the reviewer and would like to clarify the exact data splits that are used in model training and evaluations.
>
> Tasks:
> The public dataset includes multiple monkeys performing center-out (CO) and random target (RT) tasks.
>
> Train and Test Data Pipeline:
> We designed a custom preprocessing pipeline to build the train dataset (for pretraining) and test dataset (for downstream evaluation) via torchbrain. In particular, the SpikeBinner in the pipeline converts spike timestamps to binned spike counts for transformer input, each data is 1s with 20ms bins without overlapping windows, leading to (n_winodws, 50, n_units). For the pretraining dataset, we filter the data from intervals.start to interval.end (thus not trial-aligned). For the downstream evaluation dataset, we use the data from intervals.go_cue_time and interval.stop_time for CO tasks, and again use data from intervals.start to interval.end for RT tasks. We build the data via non-overlapping sequential bins per session, and there is no random shuffling of the datasets. We use the same temporal order of 70%, 10%, 20% for train/val/test data split as in POSSM.
>
> Indeed, there is no precise consistency between the evaluation of baselines performed in [1] that we include as comparison in the tables of our work as well, and RPNT(see the following output evaluation response Q8),, from a high-level perspective, both evaluations are strictly causal with the same data splits. Further, the training and evaluation for RPNT were consistently performed on non-overlapping 1s time windows. We followed a custom pipeline and evaluation, but similar to [1], since at the submission of our work, the pipeline and evaluation code details had not been released.
>
> We appreciate the remark of reviewer that the Wiener filter baseline cited from PoYo work [2], which might be set up with a random shuffling data split. We reran the Wiener filter baseline and updated the table as below.
>
> | Method | C-CO | T-CO | T-RT |
> |--------|------|------|------|
> | Wiener filter (Table 1) | 0.8712 ± 0.0137 | 0.6597 ± 0.0392 | 0.5942 ± 0.0564 |
>
> [1] Ryoo*, Krishna*, Mao*, et al. "Generalizable, real-time neural decoding with hybrid state-space models." NeurIPS 2025. URL: https://openreview.net/forum?id=1i4wNFgHDd (arXiv v1: https://arxiv.org/pdf/2506.05320v1)
>
> [2] Azabou et al. “A unified, scalable framework for neural population decoding.” NeurIPS 2023. URL: https://proceedings.neurips.cc/paper_files/paper/2023/hash/8ca113d122584f12a6727341aaf58887-Abstract-Conference.html
>
>
> **W1:Q3**
>
> We thank the reviewer for mentioning the similarity of the Full-SFT regime and  POSSM FT regime.
> Indeed, POSSM uses the FT regime definition as a regime for full model fine-tuning rather than the full downstream data fine-tuning. This is indeed analogous to the FS-SFT regime that we define in our work. We have corrected our and compared RPNT FS-SFT results with POSSM baselines only.
>
> **W1:Q4**
>
> We would like to clarify that the standard deviations in Table 1 are reported across sessions. This is consistent with POSSM. We will add additional clarifications to the caption of Table1 and also clarify that the standard deviations in Figure 3 are reported across successful trials.
>
> **W1:Q5**
>
> We use the public dataset of (Perish2018) since it is a comprehensive dataset for model generalization testing (with large amounts of self-contained data for supporting cross-session, cross-task, and cross-subject experiments). The POSSM model has been the latest model tested on it.
>
> The goal of our work is not only to surpass the benchmark accuracy of existing approaches, such as  POSSM, PoYo, or NDT-type models. Our additional and key aim in this work is to examine, from basic construction, various transformer components (besides neural tokenization) and adapt them to neural data domain-specific transformer design. Indeed, the insightful neural tokenization perspective of POSSM and PoYo advances neural motor transformer input design, while our work advances transformer block design and self-supervised neural transformer pretraining. We will revise the related work and discussion to clarify the different approaches and RPNT's unique contribution.

---

> > ### Author Response · Authors · 2025-11-28
> > **Continue**
> >
> > **W2:Q6**
> >
> > 1) RPNT does not necessarily need to be trial-aligned. For example, in RT tasks, there is no trial definition (only the start and end timepoints of the reaching sequence). The RPNT model does require a consistent data segment so that the transformer can do efficient batch-based processing. It can be 1s or 5s, depending on the application in practice.
> > 2) Automatic handling of a varying number of neurons is an important property of POSSM and PoYo. RPNT handles it differently - by resampling the neuron activity to the maximum number of neurons. For example, assume for some session, we have the data size with (T, 48), where 48 is the number of neurons, and we need it to be consistent with (T,100) for the batch process. Thus, we do random neuron sampling of another 52 neural activities to reach (T,100) e.g., (T,48) data is replicated and replicated again to repeat for the first four neurons (T, 4) to compose (T,52). Since uniform random masking is always applied in RPNT pretraining, such a simple solution to robust model pretraining. Such a strategy would apply to another session with more neurons than 100, then 100 neurons will be randomly picked to consider (T,100).
> >
> >
> > **W2:Q7**
> > In our implementation, we construct the historical data T_hist from T with temporal masking, meaning that we use up-to-date historical data as the context history for the generation of convolution kernels. Another historic data T_hist can also be constructed via receding windows, e.g., we will use the most recent K steps. Here, for simplicity, we use the first implementation.
> >
> >
> > **W2:Q8**
> > We are happy to comment on the bin sizes used in the study.
> > 1). Each input data is 1s with 20ms bins without overlapping windows, leading to (batch_size, 50, n_units).
> > 2) We follow the standard readout layer for behavior decoding, which means for each transformer input data (50, n_units), we get the output prediction (50, 2). Then, we calculate the corresponding R-squared score between our output prediction and the label (50, 2).
> >
> > **W2:Q9**
> >
> > With respect to the causal convolution kernel-based attention, our implementation applies the convolution first before the causal masking, as this ensures the temporal causality at the very end. This may slightly perturb causal training (create a leak during training) because of the kernel perception field. To strictly test this point, as the reviewer suggested, we implement the reviewer's suggestions. Specifically, we first apply a temporal masking, and then we apply the convolution operation. Below, we show the results on the T-RT and B-CS tasks (as they are harder compared to the C-CO and T-CO tasks). Our results did not show obvious performance differences in both implementations. We will incorporate these arguments and ablation studies in the revised version of the paper.
> >
> > | Method | T-RT | B-CS |
> > |--------|------|------|
> > | Temporal masking first (reviewer suggest) | 0.8474 ± 0.0902 | 0.6617 ± 0.0219 |
> > | Convolution first (our implementation) | 0.8515 ± 0.1071 | 0.6612 ± 0.0328 |
> >
> >
> > **W2:Q10**
> >
> > We design the dataset assigned numbers {0,1,2,3} for c,j,m,t according to alphabetical order. Regarding the more diverse datasets, there are two ways for scaling. 1) We can keep using the alphabetical order (say #N < 26); 2) In the appendix, we mention we can train a separate MLP encoder for string-based meta information to get the embedding. Specifically, we can follow the neural codebook design to learn a metadata codebook, which is also an interesting direction to explore.
> >
> > We performed another experiment to address the reviewer’s concern, where we randomly changed the dataset assigned numbers to {1,3,2,0} for c,j,m,t, and tested it on the T-RT test. Our results did not show obvious performance differences in different datasets assigned numbers.
> >
> > | Assigned number| T-RT |
> > |---------------|------|
> > | {1,3,2,0} | 0.8479 ± 0.0960 |
> > | {0,1,2,3} | 0.8515 ± 0.1071 |
> >
> >
> > **W2:Q11**
> >
> > The contrastive loss is proposed for neuropixel data to enforce the RPNT model with the ability to distinguish the neural embedding for different brain sites. We will include this comment in the main text.

---

> > > ### Author Response · Authors · 2025-11-28
> > > **Continue**
> > >
> > > **W2:Q12**
> > > We thank the reviewer for an interesting proposal to test transfer across datasets since both datasets are monkeys doing similar motor tasks with recording on similar regions. We designed two experiments for evaluating the transfer dataset generalization. First, we apply the neuropixel pretrained RPNT model on the public dataset. We observe that RPNT has limited generalization performance.
> > >
> > > | Method | C-CO | T-CO | T-RT |
> > > |--------|------|------|------|
> > > | Transfer from neuropixel to public | 0.8603 ± 0.0009 | 0.7585 ± 0.0344 | 0.6395 ± 0.0892 |
> > > | No transfer, only on public | 0.9894 ± 0.0037 | 0.9626 ± 0.0059 | 0.8778 ± 0.1005 |
> > >
> > > On the other hand, when we flip the transfer and apply pretrained RPNT on the public dataset transferred to the neuropixel dataset, we observe that RPNT is able to maintain good generalization performance.
> > >
> > > | Method | B-CS |
> > > |--------|------|
> > > | Transfer from public to neuropixel | 0.6537 ± 0.0413|
> > > | No transfer, only at neuropixel | 0.6612 ± 0.0328 |
> > >
> > > It is interesting to observe that the pretraining on the public dataset contains robust neural representations that can be adapted to the neuropixel datasets. This may be partially explained by the amount of data between the public dataset (more than 100 sessions) and the neuropixel dataset (16 sites/sessions).
> > >
> > > **W2:Q13**
> > >
> > > We provide more results by sweeping the train split ratio from [1%,5%,10%,20%,30%,40%,50%], and the sweeping results are shown as follows. In general, the pretrained RPNT model outperformed other baselines.
> > >
> > > | Method | 1% | 5% | 10% | 20% | 30% | 40% | 50% |
> > > |--------|------|------|------|------|------|------|------|
> > > | Wiener filter | 0.0009 | 0.0344 | 0.0892 | 0.3462 | 0.3567 | 0.3597 | 0.3634 |
> > > | MLP | 0.1181 | 0.3070 | 0.3704 | 0.4074 | 0.4215 | 0.4279 | 0.4314 |
> > > | RNN | -0.0049 | -0.0215 | 0.1083 | 0.5015 | 0.5534 | 0.6229 | 0.6295 |
> > > | NDT | -0.0038 | 0.0836 | 0.2401 | 0.5272 | 0.5752 | 0.6082 | 0.6175 |
> > > | PoYo | -0.0136 | 0.2285 | 0.5378 | 0.5944 | 0.6578 | 0.6938 | 0.7285 |
> > > | RPNT (from scratch) | -0.4101 | 0.1553 | 0.4884 | 0.6358 | 0.6883 | 0.6969 | 0.7265 |
> > > | RPNT (pretrained) | 0.0701 | 0.3735 | 0.5418 | 0.6612 | 0.6927 | 0.7170 | 0.7329 |
> > >
> > > **W2:Q13-Q16**
> > >
> > > We thank the reviewer for identifying the specified typos. We will fix them in the revised version of the manuscript.

---

### Official Review · Reviewer_vuBp · 2025-10-31

**Soundness:** 3
**Presentation:** 3
**Contribution:** 2
**Rating:** 6
**Confidence:** 3

**Summary:**

The authors present a large neural model using a transformer architecture for the purpose of decoding behavior from measured neural activity. In a self-supervised learning phase (SSL), a generalist neural encoder is pretrained to generate neural embeddings. In a subsequent  Supervised Fine Tuning phase (SFT) a readout layer is appended to the neural encoder to predict behaviour. Three main improvements on previous work are detailed. First the authors use a multidimensional version of rotary positional embeddings (MRoPE) where dimensions correspond not only to time as previously done but also to meta-data (recording positions, subject, task). Then the inherent non-stationarity of neural activity is captured using context-based attention mechanisms using convolution kernels. At last the pre-training masking strategy is not fixed but random following a uniform distribution for both time and neuron.
The model is then put to the bench on extracellular neurophysiology recordings in primates using Utah arrays (dataset A) and multiple Neuropixel insertions (dataset B). The Neuropixel dataset contains multiple sites allowing to test the spatial encoder, while both datasets provide opportunities to test the temporal encoder, multiple sessions, multiple tasks and multiple subjects. The authors present superior R2 decoding results compared baselines in a cross-session, cross-subject and cross-task scenarios for zero-shot, few-shot fine-tuning and full fine-tuning. The zero-shot model presented even matches or exceed fully fine tuned baselines.
Ablation studies show the context based attention mechanism the main driver of improvement, followed by MRoPE and then the pre-training masking strategy.

**Strengths:**

The concepts are clear and well exposed with the 3 main drivers of performance well motivated: non-stationarity of neural data, multi-dimensional meta-data encoding and the pre-training masking strategy.
The model shows substantial improvement on the baselines, even in a few shot or zero shot regime, on a relatively well-solved decoding task (R2 above 90% for existing baselines).
The ablation studies are useful for a practitioner wanting to explore further neural transformer architectures by ordering which improvement drove the performance the most.
The limitations of the model are disclosed: the model is pre-trained on a single modality on a narrow range of brain locations.

**Weaknesses:**

- Authors mention possible future BCI application l. 109, without mentioning computational requirements. It is probable that closed-loop applications will require implementations that have lower computational requirements by orders of magnitude of those large transformers. The claim is particularly dubious on a decoding problem where a simple Wiener filter can achieve a R2 around 0.88.
- Claims about insights on motor behaviour are not substantiated, here we would like to see further analysis and discussion rather than attention matrix heatmaps in supplementary figures. For example how does the attention matrix analysis outperform good old Pearson correlation analysis (or other functional correlation method) ? What are those neuroscientific insights ?
- Although the use of m-rope may be novel for large neural models, there is a similar M-Rope  implementation in vision for Qwen2 Peng Wang 2024a. URL https://doi.org/10.48550/arXiv.2409.12191. M stands for multi-modal, but the multi-dimensional block rotation matrix building follows the same pattern as presented by the authors.
- On the public benchmark, no task is "unseen", only the distribution of the task representation in the pre-training data changes. This is a bit misleading, and a proper unseen task experiment may be interesting, especially given the similarity of the protocol (reaching tasks).
- Neither the Neuropixel dataset, nor the model, nor the code are disclosed.


##### Minor comments:
Figure 1D: typo fintuneing
Figure 1C: add SSL acronym in subcaption

**Questions:**

What happens if a task is entirely removed from the pre-training set ? This means removing the behaviour component of the M-ROPE.
- What is the Neuropixel POYO / POSM baseline score ? Couldn't we expect above NDT ?
- Any insight about why the overall R2 decoding scores of Utah dataset cross-subject are so much higher than Neuropixel dataset cross-site ?

---

> ### Author Response · Authors · 2025-11-28
> **We appreciate the reviewer’s thoughtful review,feedback and support for our work. We provide  a point-to-point response to each comment below. We are happy to further clarify in case there are additional questions.**
>
> **W1**
>
> We agree that the simple Wiener filter baseline can already achieve reasonable decoding performance. Actually, this is common in the traditional Center Out (CO) decoding tasks. However, it should be noted that in future practical BCI applications, the tasks will not be limited to CO tasks. For example, the Wiener filter performance on random reaching tasks achieves only 0.59, which is not sufficient for many applications. Further, previous work [1-2] has shown that for brain-to-text or brain-to-speech decoding, more complex models, such as transformers, will be beneficial for achieving desirable performance.
>
> We have specified model hyperparameters and computational resources in Appendix J. We agree with the reviewer that deep learning, especially transformer-based models, needs heavier computational resources compared to the Wiener filter. A few ongoing works are exploring applications of post-training quantization and model pruning/distillation to allow for model compression and faster inference. We will include the discussion of these limitations and possible future directions in the revised manuscript.
>
> [1] Ryoo, Avery Hee-Woon, et al. "Generalizable, real-time neural decoding with hybrid state-space models." arXiv preprint arXiv:2506.05320 (2025).
> [2] Ye, Joel, et al. "A Generalist Intracortical Motor Decoder." bioRxiv (2025).
>
> **W2**
>
> We thank the reviewer for this comment and are happy to clarify our intent in the inclusion of neuroscientific insights. Our work is indeed focused on the computational perspective; however, we wanted to include several observed insights regarding spatial attention maps in RPNT that may lead to interpretability and connection of the attention maps with functional connectivity. RPNT attention maps are based on latent neural presentations and an outcome of a measure of causal masked neural interactions. This differs significantly from Pearson correlation or Coherence analyses that are based on signal statistics.
>
> Our intent to include RPNT attention map observations is to connect with recent studies such as [1], which demonstrate that task information is distributed spatially within the PMd-M1 brain region, revealing the heterogeneity structure in the PMd-M1 region. In RPNT spatial attention map, we observe that only a few sites are active during the reaching period, which suggests similar data-driven observations as in these studies. Further, our attention map informs the temporal evolution of the site activation, which may provide insight for future intervention investigation of brain sites within the PMd-M1 region. We will include a discussion above in the revised manuscript, clarifying the limitations of the observations of RPNT attention and possible future work needed to verify these observations further.
>
> [1] Canfield, Ryan A., et al. "The spatiotemporal structure of neural activity in the motor cortex during reaching." bioRxiv (2025): 2025-10.
>
> **W3**
>
> We are thankful to the reviewer for this comment. We will modify and extend our description of MRoPE to focus on the interpretation of multidimensions, i.e., the distinction of MRoPE from RoPE, with M being the configuration of dimensions to aggregate experimental metadata.
>
> We will also include a discussion of  MRoPE use in [1] in the related work section of positional embeddings as follows: Recent extensions of RoPE to multiple dimensions have shown promise in handling complex positional relationships beyond simple sequential ordering. The Qwen2 [1] demonstrates effective multi-dimensional RoPE for handling both sequence position and other structural dimensions. Here, we embrace RoPE for neural activity and use the extended MRoPE [1] for capturing recording configurations.
>
> [1] Wang, Peng, et al. "Qwen2-vl: Enhancing vision-language model's perception of the world at any resolution." arXiv preprint arXiv:2409.12191 (2024).
>
> **W4**
>
> Indeed, this is an insightful future experiential design. The current public dataset does not support such an “unseen task” experiment (there’s no extra reaching task category).
>
> **W5**
>
> We intend to open-source the code and model (pretrained) upon publication of the work.
> Minor Comments:
> We have updated Figure 1 according to the reviewer's suggestions.

---

> > ### Author Response · Authors · 2025-11-28
> > **Continue**
> >
> > **Q1**
> >
> > If we remove the meta-data integration in the M-RoPE design, then it will become standard RoPE, for which as we show in the ablation study of Section 4.4 and Table 2, the performance drops by approximately 3% .
> >
> > **Q2**
> >
> > We thank the reviewer for the question. We provide the result for POYO on the neuropixel data. We could not compute POSSM results since the POSSM  code has not been released.
> >
> > | Method | Cross-Site (B-CS) |
> > |--------|-------------------|
> > | Wiener filter | 0.3462 ± 0.0710 |
> > | MLP | 0.4074 ± 0.0592 |
> > | RNN (LFADS) | 0.5015 ± 0.1085 |
> > | Transformer (NDT) | 0.5272 ± 0.0720 |
> > | Transformer (PoYo) | 0.5944 ± 0.0901 |
> > | RPNT (from scratch) | 0.6358 ± 0.0311 |
> > | RPNT (pretrained) | 0.6612 ± 0.0328 |
> >
> >
> > **Q3**
> >
> > We thank the reviewer for asking about the difference between the datasets.
> >
> > The key distinction of the neuropixel dataset is that only a single probe is placed in the chamber (see Appendix I.1 Figure 2 for probe insertion locations). This creates an additional challenging generalization problem even for the same subject. In this scenario, the pre-training and the downstream data are recorded from different brain sites. A recent study [1] showed that task information is distributed spatially within the PMd-M1 brain region, revealing the heterogeneity structure in the PMd-M1 region. Thus, each cross-site recording only captures partial related neural dynamics.
> >
> > The Utah array, on the other hand,  is more consistent in terms of the neural data collection, i.e., most of the neurons’ activity will be repeatedly recorded across sessions. As a result, each session's data is almost self-contained to record the main neural-behavior related dynamics, contributing to stronger generalization ability even across subjects.
> >
> > [1] Canfield, Ryan A., et al. "The spatiotemporal structure of neural activity in the motor cortex during reaching." bioRxiv (2025): 2025-10.

---

### Official Review · Reviewer_wwYf · 2025-10-31

**Soundness:** 2
**Presentation:** 3
**Contribution:** 2
**Rating:** 2
**Confidence:** 4

**Summary:**

The paper presents new transformer model for brain-computer interface decoding of motor tasks. They demonstrate its performance on offline datasets and compare to other transformer models.

**Strengths:**

- State of the literature was summarized well in this crowded application space.

- Each model component is motivated, diagramed (Fig. 2), and explained very clearly.

- Datasets are standard for motor BCI tasks and a fair number of model comparisons were implemented.

- Limitations were discussed ("tasks represent simplified and stereotyped motor behaviors") appropriately.

**Weaknesses:**

- Related work was limited to neural decoding and did not explore relevant literature in transformers for time series, non-stationary data in general.

- The paper seems highly application-specific and driven by offline decoding results. The novelty is weakened by the combining of existing mechanisms (RoPE, masking, context-based attention) for a task-specific model rather than a fundamentally new insight into the performance of transformers for neural data applications.

- Section 3.2 on context-based attention mechanism wasn't clear on what was novel and what exists already. Attention pooling and convolutional kernels are quite common in transformer applications.

- Results do not always appear to be significantly different when selecting for best model. Distinguishing between actually different (statistically) and on average slightly better would be better.

- Offline performance of a decoder does not strictly correlated with online performance. New recordings were done but it is not clear that the model results were collected online.

- Ablation experiment results were stated without much insight into mechanisms.

**Questions:**

- How were the comparison models chosen? Why not NDT3 or BrainBERT?

- Which results were significant improvements?

- Is MRoPE is a novel generalization to multiple dimensions? The original paper covers a 2D case and a general form. The text may be misleading to state RoPE was extended to multiple dimensions here without additional qualifiers (line 202).

- Why were these tasks chosen: cross-session, cross-type, cross-subject, and cross-site? Cross-session for generalization across time, type for task, subject for if no training is possible?, site for ...? Aren't different subjects already giving different sites? Or are there different brain regions used for some reason? It seemed to be the same usual regions (PMd, M1).

---

> ### Author Response · Authors · 2025-11-28
> **We thank the reviewer for their thoughtful review and feedback. We provide a point-to-point response to each comment below and we are hopeful that these resolve reviewer concerns. We are happy to further clarify in case there are additional questions.**
>
> **W1**:
>
> We thank the reviewer for suggesting adding recent advances in transformers for time series similar to neural data. We plan to add to the related work a description of recent transformer advances in nonstationary time series applications along the lines of the following:
> Non-stationary Transformers [1] introduce a series of stationary and de-stationary attention to handle distribution shifts in time-series forecasting, while FEDformer [2] employs frequency-enhanced decomposed architectures to capture both seasonal and trend patterns in nonstationary signals. Additionally, crossformer [3] utilizes dimension-segment-wise attention to capture cross-time and cross-dimension dependencies.
> [1] Liu, Yong, et al. "Non-stationary transformers: Exploring the stationarity in time series forecasting." Advances in neural information processing systems 35 (2022)
> [2] Zhou, Tian, et al. "Fedformer: Frequency enhanced decomposed transformer for long-term series forecasting." International conference on machine learning. PMLR, 2022.
> [3] Zhang, Yunhao, and Junchi Yan. "Crossformer: Transformer utilizing cross-dimension dependency for multivariate time series forecasting." The eleventh international conference on learning representations. 2023.
>
> **W2**
>
> We are happy to clarify the novelty and the contribution of our work. Indeed, our work is application-driven and has been submitted to the applications track of ICLR. The motivation for our research is that there is a need for transformer architectures designed specifically for neural signals, capable of pretraining and generalization (not for a specific task). It appears that specific components of leading transformer architectures may or may not apply to neural signals, and thus the introduction of such components into neural signal transformers requires basic principles design studies.
>
> For example, the metadata usage, which we propose as a novel component for RPNT, was not introduced previously as far as we know, in neural data modeling, but as we show, the combination of metadata along with the well-established RoPE component contributes significantly to the enhancement of transformer performance. Our goal in this paper was to examine basic principles and identify components of general transformers that would contribute to novel components and further the efficiency of neural signals transformers. Indeed, our ablation studies show that the proposed novel components improve the decoding performance. We will incorporate the discussion above into a revised version of the manuscript.
>
> **W3**
>
> The novelty of our context-dependent attention design is as follows. a) Our convolution kernels are learnt through neural embeddings, which originally can be treated as the context from the past historical neural data. b) Essentially, this distinguishes the design in conformer[1] (existing convolution transformer paper), in which the convolution kernel is directly learnt without building the connection to contextual neural embeddings. c) From a high-level perspective, our design goal is to leverage neural data in the past to infer the context of neural latent to handle non-stationarity (i.e., let the attention focus on the local stationary part rather than the global nonstationary component). Using convolution kernels is an implementation to achieve this efficiently through the modification of attention operations.
> [1] Gulati, Anmol, et al. "Conformer: Convolution-augmented transformer for speech recognition." arXiv preprint arXiv:2005.08100 (2020).
>
> **W4**
>
> For C-CO and T-CO tasks in the public dataset, RPNT improvement is less than 5%. This is due to a) most of the training data is the center-out (CO) tasks. b) CO tasks were the most distinguishable behavior tasks in traditional motor decoding. For Cross-subject random reaching (T-RT) tasks, however, our results significantly improve the decoding performance (by more than 10%). T-RT tasks are more complex and have fewer examples in the pretraining data. Furthermore, on the neuropixel dataset, our results also outperform the SOTA more than 5%. Therefore, we claim that RPNTdesign significantly outperformed other architectures. We ran sweep studies on the RPNT model for T-RT tasks, as shown in Appendix tables 3-5,  and show that on average, our model performs consistently for a wide range of hyperparameters.
>
> **W5**
>
> We appreciate the reviewer clarifying online vs. offline validation. The results were obtained through offline analysis, which emulates an online regime, i.e. causal regime. A true online performance estimation requires recording of neural data and synchronized application of RPNT in real time, which includes multiple hardware and software challenges yet to be solved. These are not within the scope of this work. This work focuses on identifying a robust design for components to enable such an application in the future. We will include a discussion of offline vs. online evaluation in the discussion of limitations.

---

> ### Author Response · Authors · 2025-11-28
> **Continue**
>
> **W6**
>
> Due to the page limit of ICLR, we included  a shorter version (highlighting the various ablation studies that we have performed. Here, we provide a full version that we intend to include in the manuscript upon publication.
>
> ## Ablation Studies
>
> **Table 1: Ablation study of position encoding methods**
> | Method | Cross-Task (T-RT) | Cross-Site (B-CS) |
> |--------|-------------------|-------------------|
> | Sinusoidal PE | 0.8260 ± 0.0894 | 0.6242 ± 0.0267 |
> | RoPE | 0.8226 ± 0.0941 | 0.6484 ± 0.0074 |
> | Learnable PE | 0.8305 ± 0.0917 | 0.6273 ± 0.0398 |
> | **MRoPE** | **0.8515** ± 0.1071 | **0.6612** ± 0.0328 |
>
> **Table 2: Ablation study on attention mechanisms**
> | Method | Cross-Task (T-RT) | Cross-Site (B-CS) |
> |--------|-------------------|-------------------|
> | Standard attention | 0.8024 ± 0.0896 | 0.5024 ± 0.0388 |
> | **Context-based attention** | **0.8515** ± 0.1071 | **0.6612** ± 0.0328 |
>
> **Table 3: Ablation study on masking strategies during pretraining**
> | Masking Strategy | Neuron ratio | Temporal ratio | Cross-Task (T-RT) | Cross-Site (B-CS) |
> |------------------|--------------|----------------|-------------------|-------------------|
> | Fixed | 0.25 | 0.25 | 0.8349 ± 0.0948 | 0.6549 ± 0.0254 |
> | Fixed | 0.50 | 0.50 | 0.8437 ± 0.0968 | 0.6557 ± 0.0316 |
> | Fixed | 0.75 | 0.75 | 0.8392 ± 0.0975 | 0.6593 ± 0.0251 |
> | Fixed | 0.25 | 0.75 | 0.8414 ± 0.0990 | 0.6540 ± 0.0279 |
> | Fixed | 0.75 | 0.25 | 0.8398 ± 0.0922 | 0.6594 ± 0.0301 |
> | **Random** | **U(0, 1)** | **U(0, 1)** | **0.8515** ± 0.1071 | **0.6612** ± 0.0328 |
>
> To validate the effective components of RPNT, we conducted ablation studies using the most challenging T-RT task and the novel B-CS task under the FS-SFT setup. Table 1, Table 2, and Table 3 summarize the results for position encoding, attention mechanism, and masking strategy, respectively. We highlight the outcomes of the studies in the following paragraphs.
>
> **MRoPE.** We compared MRoPE against three alternative positional encoding approaches (see appendix): sinusoidal PE, standard RoPE, and multi-dimensional learnable PE in Table 1. MRoPE achieved the highest R² score of 0.8515 ± 0.1071, demonstrating consistent improvements over sinusoidal PE (0.8260 ± 0.0894), standard RoPE (0.8226 ± 0.0941), and learnable PE (0.8305 ± 0.0917). Approximately 3% improvement over standard RoPE and 2% improvement over multi-dimensional learnable PE validated our design of extending one-dimensional RoPE to multiple dimensions, enabling better generalization to session configurations while preserving relative position information.
>
> **Context-based Attention Mechanism.** Table 2 demonstrates the importance of context-dependent attention design. Replacing the proposed context-based attention with standard self-attention resulted in a substantial performance drop from 0.8515 ± 0.1071 to 0.8024 ± 0.0896 (approximately 5% relative decrease). This significant gap highlights the effectiveness of the adaptive kernel generation mechanism that we proposed in handling the non-stationary nature of neural recordings (see appendix for kernel size sweeping study).
>
> **Random Uniform Masking Strategy.** Fixed masking ratios, whether symmetric or asymmetric across neuron and temporal dimensions, consistently underperformed the random uniform masking approach that we propose for RPNT (see Table 3). This result showed that exposing the model to the full spectrum of masking ratios led to robust neural representations with better downstream performance.
>
>
> **Q1**
>
> 1) We chose to compare with recently published public benchmarking results performed in [1]. Unless specifically noted in the manuscript, these results were reported in the associated publication [1]. As the publication uses NDT2 rather than NDT3, we keep using NDT2 for consistency.  We thank the reviewer for the question. In the revised manuscript, we will make sure to clearly refer to [1] in tables that use these results for comparison.
>
> 2) Brainbert has similarities with RPNT in being a self-supervised pre-training model. However, Brainbert is a non-causal model based on the stacking of transformer encoder layers, which results in bidirectional attention. In contrast, RPNT is causal and applies transformer decoder design and training with causal masked neural modeling.
>
> [1] Ryoo*, Krishna*, Mao*, et al. "Generalizable, real-time neural decoding with hybrid state-space models." NeurIPS 2025. URL: https://openreview.net/forum?id=1i4wNFgHDd (arXiv v1: https://arxiv.org/pdf/2506.05320v1)
>
>
> **Q2**
>
> Significant improvements were in the T-RT (public) task and the B-CS (neuropixel) dataset. We provided further details in the response to W4.

---

> ### Author Response · Authors · 2025-11-28
> **Continue**
>
> **Q3**
>
> We are thankful to the reviewer for this observation. We will modify and extend our description of MRoPE to focus on the interpretation of multidimensions, i.e., the distinction of MRoPE from RoPE, with M being  the configuration of dimension to aggregate experimental metadata. We will also include the MRoPE in our related work section of the positional embeddings.
>
> **Q4**
>
>  The four tasks were chosen since the first three tasks (cross-session, cross-task, cross-subject) were considered in prior works for evaluating model generalization (Page 6 experimental setup in [1]). We further propose a fourth, cross-site task, for the supplementary neuropixel dataset. In the neuropixel dataset, only a single probe is placed in the chamber (see Appendix I.1 Figure 2 for probe insertion locations). This creates an additional challenging generalization problem even for the same subject. In this scenario, the pre-training and the downstream data are recorded from different brain sites. A recent study [2] showed that task information is distributed spatially within the PMd-M1 brain region, revealing the heterogeneity structure in the PMd-M1 region. As the neuropixel technology is new and recordings are limited, our work is also one of the few studies that considers the generalization of neural transformers with respect to neuropixel modality.
>
> [1] Ryoo*, Krishna*, Mao*, et al. "Generalizable, real-time neural decoding with hybrid state-space models." NeurIPS 2025. URL: https://openreview.net/forum?id=1i4wNFgHDd (arXiv v1: https://arxiv.org/pdf/2506.05320v1)
>
> [2] Canfield, Ryan A., et al. "The spatiotemporal structure of neural activity in the motor cortex during reaching." bioRxiv (2025): 2025-10.

---

### Public Comment · ~Nanda_H_Krishna1 · 2025-11-17
**Pointing out key issues and inaccuracies in baseline details and comparisons**

Dear authors and reviewers,

I am a lead author of POSSM [1], a paper cited in this submission and one of the key baseline methods that RPNT is compared to. I would like to thank the authors for considering our work for comparison. However, there are several key *inconsistencies, inaccuracies, and lack of clarity* that I am unfortunately compelled to point out in this submission, some of which were keenly raised by [Reviewer sU3Q](https://openreview.net/forum?id=NokbiILX9n&noteId=BHJft7TbIj). I provide a list of these points below and hope the authors will correct or clarify them.

### **1. Reporting the POSSM and POYO papers' baselines without proper attribution**

The values in Table 1 reported in this work is an **exact copy of Table 1 from the [POSSM arXiv v1 preprint’s](https://arxiv.org/pdf/2506.05320v1) Table 1**, with **2 key typos**, i.e., missing leading zeros after the decimal point (o-POSSM-Mamba and o-POSSM-GRU standard deviations on T-RT). However, the paper has **neither properly credited us for our work** in running all these baselines (MLP, S4D, Mamba, GRU, POYO, POSSM, NDT-2, POYO-1, o-POSSM) **nor has it mentioned that the results are taken from the POSSM paper** in the caption or Appendix F. The results from our Table 1 have also been combined with the Wiener filter results **exactly taken from the POYO paper’s Table 2** [2], which is inconsistent as I will discuss below.

### **2. Using different, inconsistent data splits across baselines and RPNT**

The results in this paper **inconsistently compare** with POSSM, POYO, and our baselines by **using different data splits**. The authors mention that they use the `brainsets` split for their experiments — however this makes **Figure 1 showing the data splits incorrect** — the **split in `brainsets` is shuffled** within and not causal with respect to time in each session. Meanwhile, the **POSSM paper uses a causal split** as indicated by POSSM’s Appendix A Figure 5. The POYO paper from which the Wiener filter results were taken uses yet another split, making this comparison inconsistent yet again.

### **3. Inaccurately reporting POSSM and POYO as Full-SFT as opposed to FS-SFT**

The paper **inaccurately reports** that POYO-1 and o-POSSM are fine-tuned on all downstream sessions, which the authors denote as Full-SFT. This is **incorrect** – **POYO-1 and o-POSSM are only fine-tuned on single downstream sessions separately**, making it identical to the **FS-SFT** regime mentioned here. The standard deviations in Table 1 for our methods are reported across sessions, as clearly mentioned in POSSM’s Table 1 caption. It is also unclear what the authors report their standard deviations over.

### **4. Unclear whether models are compared on same number of output predictions**

We evaluated POYO, POSSM and other baselines on **all behavioural outputs present in the session/dataset**. That is, even with a bin size of 50ms, POSSM and POYO were evaluated on **all outputs** within that 50ms due to the flexible output module. In the Perich et al. dataset, this would amount to 5 outputs per 50ms bin. However, the RPNT model’s outputs follow a simple linear readout scheme based on details in the paper, so it is **unclear if RPNT was evaluated on the same outputs, and same number of outputs**, i.e.,  just one output per 50ms bin, or even an average-pooled version of all outputs per bin – **both of which would be inconsistent with our work** and unfair to us as we predict have to accurately predict more outputs.

### **5. Carefully ensuring causality**

We followed a careful, **causal evaluation** protocol for all baselines including POYO in our paper. This meant that POYO would only predict behaviour associated with the final 50ms of each 1s context window, and the window would have a stride of 50ms to ensure prediction on all outputs. It is **currently unclear** whether the architecture and evaluation of RPNT is actually causal, i.e., whether the “causal” contextual attention variant is actually causal since the convolution is applied before causal masking. Thus, it is unclear from the available details in the paper whether causality is actually maintained or violated, and this should be ensured for a fair comparison with our work.

---

> ### Public Comment · ~Nanda_H_Krishna1 · 2025-11-17
> **Pointing out key issues and inaccuracies in baseline details and comparisons (contd.)**
>
> Finally, we would like to also point out key differences between our models and the proposed RPNT model: POSSM and POYO do not require any trial-alignment and are robust to differing numbers of neurons across sessions, also allowing for efficient finetuning by learning just neuron embeddings for new downstream sessions. However, RPNT requires trial alignment, and it is unclear how the method deals with varying numbers of neurons across sessions or how it would be amenable to efficient finetuning to new sessions. We hope these key differences will be acknowledged.
>
> Overall, we hope that the authors will take these points into account and clarify or correct these issues. We are in the process of publicly releasing our code with the `torch_brain` package as part of our [NeurIPS camera ready submission for POSSM](https://openreview.net/forum?id=1i4wNFgHDd) and would be willing to share implementation details and code with the authors.
>
> **References:**
>
> 1. Ryoo*, Krishna*, Mao*, et al. "Generalizable, real-time neural decoding with hybrid state-space models." NeurIPS 2025. URL: https://openreview.net/forum?id=1i4wNFgHDd (arXiv v1: https://arxiv.org/pdf/2506.05320v1)
> 2. Azabou et al. “A unified, scalable framework for neural population decoding.” NeurIPS 2023. URL: https://proceedings.neurips.cc/paper_files/paper/2023/hash/8ca113d122584f12a6727341aaf58887-Abstract-Conference.html

---

> ### Author Response · Authors · 2025-11-28
> **We are glad to receive a response from POSSM work authors . We note that we have addressed many of the points in our response to reviewer sU3Q Here, for the rebuttal self-consistently, we also provide point-to-point responses for each comment below.**
>
> **Q1**
>
> 1) We have corrected the typos in reporting the results published in POSSM arXiv v1 preprint’s Table 1. In the revised version of our manuscript, we will have a clear citation to indicate the origin of the paper results. We will also include all the baselines for completeness in the final submission version. The submitted manuscript includes the top baselines due to the page limit.
>
> 2) In Appendix F, we mention the POSSM paper without explicitly stating that the results are taken from the POSSM paper. Now, we include the acknowledgement accordingly.
>
> 3) We remove the Wiener filter results as it have inconsistent data splits. We run the Wiener filter locally and provide the correct Wiener filter results (see below).
>
> | Method | C-CO | T-CO | T-RT |
> |--------|------|------|------|
> | Wiener filter (Table 1) | 0.8712 ± 0.0137 | 0.6597 ± 0.0392 | 0.5942 ± 0.0564 |
>
>
> **Q2**
>
> We appreciate the comment by the reviewer and would like to clarify the exact data splits that are used in model training and evaluations.
>
> Tasks:
> The public dataset includes multiple monkeys performing center-out (CO) and random target (RT) tasks.
>
> Train and Test Data Pipeline:
> We designed a custom preprocessing pipeline to build the train dataset (for pretraining) and test dataset (for downstream evaluation) via torchbrain. In particular, the SpikeBinner in the pipeline converts spike timestamps to binned spike counts for transformer input, each data is 1s with 20ms bins without overlapping windows, leading to (n_winodws, 50, n_units). For the pretraining dataset, we filter the data from intervals.start to interval.end (thus not trial-aligned). For the downstream evaluation dataset, we use the data from intervals.go_cue_time and interval.stop_time for CO tasks, and again use data from intervals.start to interval.end for RT tasks. We build the data via non-overlapping sequential bins per session, and there is no random shuffling of the datasets. We use the same temporal order of 70%, 10%, 20% for train/val/test data split as in POSSM.
>
> Indeed, there is no precise consistency between the evaluation of baselines performed in [1] that we include as comparison in the tables of our work as well, and RPNT(see the following output evaluation response Q8 in reviewer sU3Q),, from a high-level perspective, both evaluations are strictly causal with the same data splits. Further, the training and evaluation for RPNT were consistently performed on non-overlapping 1s time windows. We followed a custom pipeline and evaluation but similar to [1] since at the submission of our work, the pipeline and evaluation code details had not been released.
>
> We appreciate the remark of reviewer's remark that the Wiener filter baseline cited from PoYo work [2], which might be set up with a random shuffling of the data split. We reran the Wiener filter baseline and updated the table (see Q1 tables)
>
> [1] Ryoo*, Krishna*, Mao*, et al. "Generalizable, real-time neural decoding with hybrid state-space models." NeurIPS 2025. URL: https://openreview.net/forum?id=1i4wNFgHDd (arXiv v1: https://arxiv.org/pdf/2506.05320v1)
>
> [2] Azabou et al. “A unified, scalable framework for neural population decoding.” NeurIPS 2023. URL: https://proceedings.neurips.cc/paper_files/paper/2023/hash/8ca113d122584f12a6727341aaf58887-Abstract-Conference.html
>
>
> **Q3**
>
> 1) We thank the reviewer for mentioning the similarity of the Full-SFT regime and the POSSM FT regime.
> Indeed, POSSM uses the FT regime definition as a regime for full model fine-tuning rather than the full downstream data fine-tuning. This is indeed analogous to the FS-SFT regime that we define in our work. We will correct it in our updated manuscript and compare RPNT FS-SFT results with POSSM baselines only.
>
> 2) We would like to clarify that the standard deviations in Table 1 are reported across sessions. This is consistent with POSSM. We will add additional clarifications to the caption of Table 1.

---

> ### Author Response · Authors · 2025-11-28
> **Continue**
>
> **Q4**
>
> We are happy to comment on the bin sizes used in the study:
>
> 1) Each input data is 1s with 20ms bins without overlapping windows, leading to (batch_size, 50, n_units).
>
> 2) We follow the standard readout layer for behavior decoding, which means for each transformer input data (50, n_units), we get the output prediction (50, 2). Then, we calculate the corresponding R-squared score between our output prediction and the label (50, 2).
>
>
> **Q5**
>
>
> We clarify the following points.
>
>
> There is no data shuffling in the dataset construction as we describe in Q3.
> With respect to the causal convolution kernel-based attention, our implementation applies the convolution first before the causal masking as this ensures the temporal causality at the very end. This may slightly perturb causal training (create a leak during training) because of the kernel perception field. To strictly test this point, as the reviewer suggested, we implement the reviewer's suggestions.
> Specifically, we first apply a temporal masking, and then we apply the convolution operation. Below, we show the results on the T-RT and B-CS tasks (as they are harder compared to the C-CO and T-CO tasks). Our results did not show obvious performance differences in both implementations. We will incorporate these arguments and ablation studies in the revised version of the paper.
>
>
> | Method | T-RT | B-CS |
> |--------|------|------|
> | Temporal masking first (reviewer suggest) | 0.8474 ± 0.0902 | 0.6617 ± 0.0219 |
> | Convolution first (our implementation) | 0.8515 ± 0.1071 | 0.6612 ± 0.0328 |
>
>
> We use a causal masking strategy in our model pretraining to ensure the masked reconstruction only based on the previous time steps.
>
>
>
>
>
>
> **Q6**
>
>
> RPNT does not necessarily need to be trial-aligned. For example, in RT tasks, there is no trial definition (only the start and end timepoints of the reaching sequence). The RPNT model does require a consistent data segment so that the transformer can do efficient batch-based processing. It can be 1s or 5s, depending on the application in practice.
>
>
> Automatic handling of a varying number of neurons is an important property of POSSM and PoYo. RPNT handles it differently - by resampling the neuron activity to the maximum number of neurons. For example, assume for some session, we have the data size with (T, 48), where 48 is the number of neurons, and we need it to be consistent with (T,100) for the batch process. Thus, we do random neuron sampling of another 52 neural activities to reach (T,100) e.g., (T,48) data is replicated and replicated again to repeat for the first four neurons (T, 4) to compose (T,52). Since uniform random masking is always applied in RPNT pretraining, such a simple solution to robust model pretraining. Such a strategy would apply to another session with more neurons than 100; then, 100 neurons will be randomly picked to consider (T,100).
>
>
> The goal of our work is not only to surpass the benchmark accuracy of existing approaches such as  POSSM, PoYo, or NDT-type models. Our additional and key aim in this work is to examine, from basic construction, various transformer components (besides neural tokenization) and adapt them to neural data domain-specific transformer design. Indeed, the insightful neural tokenization perspective of POSSM and PoYo advances neural motor transformer input design, while our work advances transformer block design and self-supervised neural transformer pretraining. We will revise the related work and discussion to clarify the different approaches and RPNT's unique contribution.
>
>
>  We are glad to see and compare our model once the POSSM integrated torch_brain package is released. Also, we are eager to see if our proposed robust component can be integrated into POSSM to improve the performance in the future.

---

### Author Response · Authors · 2025-12-02
**Summary of response to all reviewers and ACs**

We sincerely thank both ICLR reviewers and the public reviewer for their helpful comments. First, we are glad that all reviewers find the merit and agree with the contribution about the innovative components in our RNPT model. Second, we have addressed the reviewer comments by adding 7 new experiments, updating tables, and expanding the text. Accordingly, we summarize the following major points in our rebuttal:

1) Regarding the related work and relevant literature review, we add sentences in the related work section to describe the recent advances in nonstationary transformer models and multimodal positional embeddings.

2) Regarding the potential data splits and unclear baseline comparison, we clarify the preprocessing details in the public dataset and ensure the comparison with our RPNT results is fair.

3) Regarding the model architecture, input/output, and downstream fine-tuning process, we clarify the technical details about the model forward/backward process in our RPNT model.

4) Regarding the extra validation of the RNPT model in generalized motor decoding, we add 7 comprehensive experiments to provide evidence of our RPNT model's generalization ability in motor decoding.

5) Regarding the limitations and future directions, we add sentences in the discussion section to describe the current limitations and potential future works.

We believe that the following rebuttal addresses all the comments, and our manuscript is further improved.

---

### Meta-Review · Area_Chair_BTpX · 2025-12-17

**Summary:**

The reviewers raised significant concerns regarding the fairness and validity of the baseline comparisons, noting that the results for competing models like POYO and POSSM appear to use different data splits and evaluation protocols than the proposed method. This same issue was raised in a public comment by one of the authors of these papers.

There was also skepticism amongst the reviewers about the model's novelty, with critics viewing it as a combination of existing techniques rather than a fundamental advance, and questioning the causal claim given the use of 2D convolutions on attention matrices. Technical clarity was anothe issue raised, as key details regarding preprocessing for variable neuron counts, bin sizes, and the precise definition of historical data were missing.

Furthermore, the reviewers doubted the practical applicability of the work, emphasizing that offline decoding success does not guarantee online BCI performance and that the high computational cost of transformers may be prohibitive for real-time use. The scope was considered too narrow, limited to monkey motor tasks without demonstrating the scalability expected of a neurofoundation model or generalization to other species. Finally, the implementation of multidimensional RoPE was criticized for introducing arbitrary inductive biases, such as the manual ordering of subjects, which may not scale or generalize well.

**Reviewer Concerns:**

I believe that the authors did successfully address some concerns about the generalizability and technical clarity. However, I'm not sure that the concerns about the appropriateness of the comparisons to past models, nor the questions around the novelty of the technique to an extent that would have satisfied the reviewers.

**Reviewer Scores:**

The original scores were 2,6,2,4. Given the partial success in addressing the reviewer concerns, I would expect partial mocves in the scores, possibly to something like 3,6,2,5, which would still put the paper just below the acceptance threshold.

---

### Decision · Program_Chairs · 2026-01-26

Reject